# Transcriptomic profiling of tissue environments critical for post-embryonic patterning and morphogenesis of zebrafish skin

Andrew J Aman[1]*[†], Lauren M Saunders[2†], August A Carr[1], Sanjay Srivatasan[2], Colten Eberhard[3], Blake Carrington[3], Dawn Watkins-Chow[3], William J Pavan[3], Cole Trapnell[2], David M Parichy[1,4]*

[1]Department of Biology, University of Virginia, Charlottesville, United States; [2]Department of Genome Sciences, University of Washington, Seattle, United States; [3]National Human Genome Research Institute, National Institutes of Health, Bethesda, United States; [4]Department of Cell Biology, University of Virginia, Charlottesville, United States

*For correspondence:
aja5x@virginia.edu (AJA);
dparichy@virginia.edu (DMP)

†These authors contributed equally to this work

Competing interest: The authors declare that no competing interests exist.

**Abstract** Pigment patterns and skin appendages are prominent features of vertebrate skin. In zebrafish, regularly patterned pigment stripes and an array of calcified scales form simultaneously in the skin during post-embryonic development. Understanding the mechanisms that regulate stripe patterning and scale morphogenesis may lead to the discovery of fundamental mechanisms that govern the development of animal form. To learn about cell types and signaling interactions that govern skin patterning and morphogenesis, we generated and analyzed single-cell transcriptomes of skin from wild-type fish as well as fish having genetic or transgenically induced defects in squamation or pigmentation. These data reveal a previously undescribed population of epidermal cells that express transcripts encoding enamel matrix proteins, suggest hormonal control of epithelial–mesenchymal signaling, clarify the signaling network that governs scale papillae development, and identify a critical role for the hypodermis in supporting pigment cell development. Additionally, these comprehensive single-cell transcriptomic data representing skin phenotypes of biomedical relevance should provide a useful resource for accelerating the discovery of mechanisms that govern skin development and homeostasis.

## eLife assessment

This study provides a clearly presented and thoughtfully analyzed single cell-resolution dataset of gene expression in wildtype and mutant zebrafish skin. These data are used by the authors to develop and test hypotheses about cell lineage relationships and signaling interactions between cell types in the skin, allowing them to identify roles for several signaling pathways and the hypodermis in scale and pigment cell development. These findings constitute a **fundamental** contribution to the field, and the rigor of the analyses make this manuscript **compelling**.

## Introduction

The steady-state chemistry of life on earth occurs within compartments bounded from the rest of the cosmos. Providing this boundary function and serving as the primary interface between organisms and their environments are sophisticated integuments that, in vertebrates, comprise marvelous and

varied skins, decorated with patterns of pigmentation and arrayed appendages including feathers, fur, or scales. Understanding the mechanistic underpinnings of animal form and phenotypic diversity is an enduring goal of basic biology, and studying skin patterning and morphogenesis can advance that goal. Additionally, while human skin is a major contributor to our outward appearance, bears all our physical interactions, and detects all our tactile sensations, it remains a failure-prone organ system with numerous poorly understood and debilitating pathologies.

Studying the skin of research organisms chosen based on phylogeny or experimental exigency can improve our understanding of regulatory mechanisms underlying integumental patterning and morphogenesis. Comparing developmental mechanisms across species can provide clues to the origin and evolution of this important organ system and may also reveal the fundamental mechanisms relevant to human health and disease. To these ends, the development of skin, and cell types within the skin, has been studied across a variety of research organisms, yielding insights into both general and species-specific mechanisms (*Duverger and Morasso, 2009*; *Chen et al., 2015*; *Patterson and Parichy, 2019*; *Aman and Parichy, 2020*).

Zebrafish (*Danio rerio*) is an outstanding research organism for studying vertebrate skin patterning and morphogenesis. Zebrafish skin, like all vertebrate skin, has a superficial epidermis composed of ectoderm-derived epithelial cells and an underlying dermis composed of mesoderm-derived mesenchymal cells and collagenous stromal matrix (*Le Guellec et al., 2004*; *Aman and Parichy, 2020*). During post-embryonic development, zebrafish skin simultaneously develops arrays of calcified scales and pigmented stripes. Both form superficially on the surface of the animal and are dispensable for survival in the laboratory, making them readily amenable to imaging and experimental perturbation, and enabling analyses of underlying cellular dynamics and molecular mechanisms (*Lee and Kimelman, 2002*; *Aman et al., 2018*; *Cox et al., 2018*; *Iwasaki et al., 2018*; *Rasmussen et al., 2018*; *Patterson and Parichy, 2019*; *Daane et al., 2016*).

To reveal potentially rare cell populations important for zebrafish scale development and pigment patterning, we used unbiased single-cell transcriptional profiling and live imaging of skins undergoing post-embryonic morphogenesis. Additionally, to gain insights into the molecular mechanisms underlying human skin pathologies, we profiled skins from *ectodysplasin a* (*eda*) mutants, *basonuclin 2* (*bnc2*) mutants, and hypothyroid fish (hypoTH; *Figure 1A*). Eda-Edar-NF-κB is a conserved signaling pathway that is necessary for normal skin appendage development in all vertebrates examined to date (*Kere et al., 1996*; *Srivastava et al., 1997*; *Kondo et al., 2001*; *Houghton et al., 2005*; *Harris et al., 2008*; *Di-Poï and Milinkovitch, 2016*). Mutations in the signaling ligand Eda-A (Ectodysplasin-A), its receptor Edar, or downstream signal transduction molecules in the NF-κB (nuclear factor-κB) pathway underlie human ectodermal dysplasias, hereditary disorders defined by loss of skin appendages and teeth (*Chu et al., 2018*). Similarly, *eda* mutant zebrafish completely lack scales, though specific mechanisms linking Eda signaling to scale formation remain unclear (*Harris et al., 2008*). To learn more about the potential interactions between pigment cells and their skin microenvironment, we profiled a mutant for *bnc2*, a conserved zinc finger containing protein implicated in human pigment variation that acts through the tissue environment to promote pigment cell development in zebrafish (*Lang et al., 2009*; *Patterson and Parichy, 2013*; *Visser et al., 2014*; *Endo et al., 2018*; *Ayoola et al., 2021*). Finally, we profiled skins from hypothyroid fish (hypoTH) that are unable to synthesize thyroid hormone (TH) owing to transgene-mediated ablation of the thyroid gland (*McMenamin et al., 2014*). TH is a potent regulator of vertebrate skin development, and thyroid dysfunction underlies debilitating skin pathologies (*Mancino et al., 2021*). We have shown that TH is necessary for dermal morphogenesis as well as pigment cell maturation and pattern formation though the underlying mechanisms remain elusive (*McMenamin et al., 2014*; *Saunders et al., 2019*; *Aman et al., 2021*).

Our analyses, using single-cell transcriptomics supplemented by histological analyses of gene expression, fate mapping, and experimental manipulations, revealed a previously undescribed epidermal cell type that expresses transcripts encoding enamel matrix proteins, relevant to understanding the ancient origins of calcified tissues, and have clarified the position of Eda within the signaling network that governs scale papilla induction. We also discovered a novel regulatory pathway that connects globally circulating TH to local epithelial–mesenchymal interactions during dermal development. We further identify the hypodermis as a crucial pigment cell supporting tissue that provides a permissive environment for the self-organizing interactions of adult stripe formation. Lastly, by comparing analyses of single-cell transcriptomes with spatial analyses of gene expression and cell

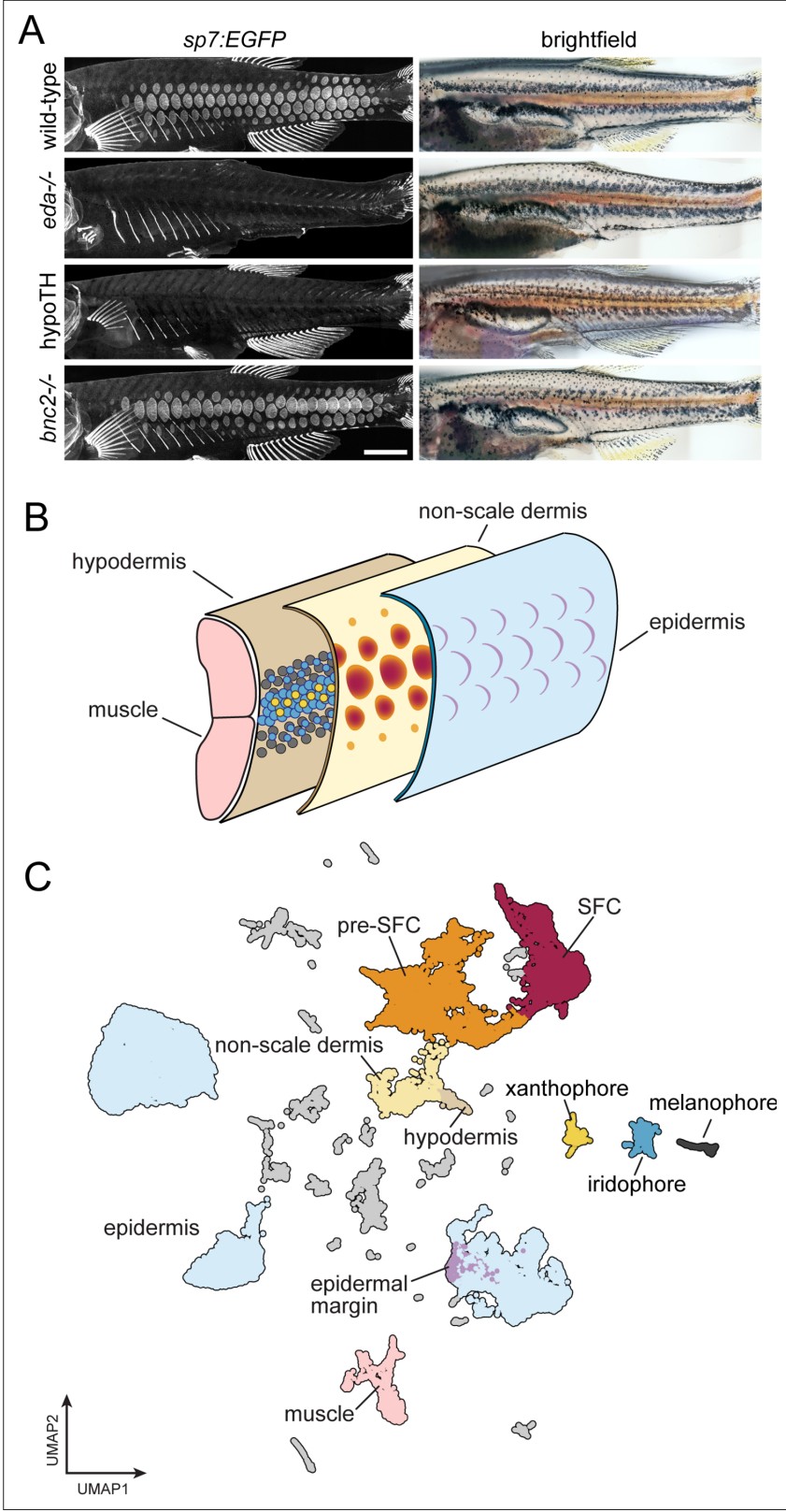

**Figure 1.** A whole-skin single-cell transcriptome from zebrafish undergoing skin patterning. (**A**) Confocal images illustrate scale-forming cells (SFCs) expressing *sp7:EGFP*, and brightfield images of the same fish show pigment pattern. At 9.2 standardized standard length (SSL), approximately 3 wk post fertilization under standard conditions, wild-type zebrafish have all steps of scale development represented and are developing a pigment pattern of

*Figure 1 continued on next page*

*Figure 1 continued*

dark stripes, with melanophores and sparse iridophores, alternating with light interstripes of densely packed iridophores and yellow xanthophores. Wild-type individuals of 9.6 SSL having precisely four rows of scales were selected for preparation of nuclei to be used in single-cell indexed RNA-seq (sci-RNA-seq); *eda* mutants and hypoTH fish, devoid of scales, as well as *bnc2* mutants having fewer, dysmorphic scales, were likewise reared to 9.6 SSL for isolation of nuclei. (**B**) Schematic representation of zebrafish skin at 9.2 SSL. The outermost layer of skin is the epidermis (blue), which develops crescent-shaped placodes (epidermal margin, lavender) above each scale. In the dermis (yellow), SFCs differentiate (orange → red). In the hypodermis (brown), dark melanophores, yellow xanthophores, and iridescent iridophores (gray, yellow, and blue circles, respectively) organize into alternating stripes. (**C**) UMAP visualization of 35,114 transcriptomes from nuclei of wild-type fish, colored by cell type assignments inferred by marker gene enrichment. Colors correspond to the schematic in (**B**). Scale bar, 500 mm (**A**).

The online version of this article includes the following figure supplement(s) for figure 1:

**Figure supplement 1.** Post-embryonic skin morphogenesis.

**Figure supplement 2.** Sci-RNA-seq data quality metrics across backgrounds.

**Figure supplement 3.** A transcriptome atlas identifies major cell types in post-embryonic skin.

**Figure supplement 4.** Unsupervised Leiden clustering of wild-type skin cells.

type differentiation, we uncover instances in which single-cell bioinformatic inferences represent, and fail to represent, true cell state transformations. Together, these analyses provide new insights into the development and evolution of vertebrate skin and highlight the importance of validating inferences of differentiation and lineage from single-cell transcriptomics with paradigms for assessing developmental events in vivo.

## Results

### sci-RNA-seq of whole skin reveals cell type diversity during post-embryonic development

Skin is a large and complex organ system, with contributions from multiple embryonic germ layers and a variety of distinct cell types. Likely due to a shared requirement for TH, pigment pattern formation and squamation occur simultaneously during post-embryonic development in different layers of the skin (*Figure 1B*; *McMenamin et al., 2014*; *Saunders et al., 2019*; *Aman et al., 2021*). To capture individual transcriptomes from a minimally biased sampling of skin cells, we performed single-nucleus combinatorial indexing (sci)-RNA-seq (*Cao et al., 2017*; *Cao et al., 2019*) on nuclei from pooled, fresh-frozen whole skins at 9.6 mm standardized standard length (9.6 SSL; *Parichy et al., 2009*), a key developmental stage of skin patterning and morphogenesis. In 9.2 SSL wild-type fish, all steps of scale morphogenesis are represented and a 'primary' pigment pattern is apparent with secondary pattern elements just beginning to form (*Figure 1A and B*, *Figure 1—figure supplement 1*, *Saunders et al., 2019*; *Parichy et al., 2009*).

To better understand the contributions of individual cell types and key factors involved in the major skin patterning events at this stage, we included three additional backgrounds representing distinct developmental perturbations: *eda* mutants, hypoTH fish, and *bnc2* mutants (*Harris et al., 2008*; *Lang et al., 2009*; *McMenamin et al., 2014*). We processed all tissue in a single sci-RNA-seq experiment, barcoding each genotype by reverse transcription index during library preparation. In total, we recovered high-quality transcriptomes from 144,466 individual nuclei with an average of 1300 unique molecular identifiers (UMIs) and 720 genes detected per cell (~60% duplication rate). Cell recovery, UMIs per cell, and numbers of genes detected were consistent across all sample groups (*Figure 1—figure supplement 2*). We removed likely multiplets (11%) using Scrublet (*Wolock et al., 2019*) and further processed the data and performed dimensionality reduction and clustering with monocle3 (*Cao et al., 2019*). To characterize cell types and developmental trajectories, we focused on the wild-type data alone (35,114 cells). We classified cells into major cell types by assessing expression of published markers for different skin and skin-associated cell types (*Figure 1C*, *Figure 1—figure supplement 3*, *Supplementary file 1*—Table 1,; *Supplementary file 2*—Table 2). The majority of cells were from epidermal and dermal populations, where we recovered transcripts from relatively rare subsets, including the edn3b/bnc2+ hypodermal monolayer that forms the deep limit of the dermis

and shha+ epidermal placode cells at the posterior scale margin (*Sire and Akimenko, 2004*; *Lang et al., 2009*; *Brock et al., 2019*; *Patterson and Parichy, 2013*; *Aman et al., 2018*). It is likely that additional cell-state heterogeneity exists within these major cell types beyond what we have annotated here (*Figure 1—figure supplement 4*, *Supplementary file 2*—Table 5). As expected, based on our stage selection, we recovered dermal scale forming cells (SFCs) and their progenitors (pre-SFCs) (*Figure 1—figure supplement 1*). We also recovered less abundant cell types, including pigment cells, lateral line cells, goblet cells, ionocytes, glia, and immune cells (*Figure 1C*, *Figure 1—figure supplement 3*).

## Decoupling of transcriptional dynamics and inferred lineage relationships during scale development and epidermal maturation

One goal of our study was to resolve transcriptional dynamics during differentiation of cell lineages that underlie skin patterning and morphogenesis. Adult zebrafish are adorned with a full coat of partially overlapping elasmoid scales, thin plates of calcified ECM that grow between the dermis and epidermis during post-embryonic development (*Figure 1*, *Figure 1—figure supplement 1*; *Sire et al., 1997a*; *Aman et al., 2018*). Scale development is associated with a population of dermal cells—referred to here as SFCs—that express genes encoding transcription factors, including *sp7* (also known as osterix, osx) and *runx2a/b*, that are necessary for differentiation of osteoblasts and odontoblasts (*Sire et al., 1997a*; *Komori, 2010*; *Li et al., 2011a*; *Zhang, 2012*; *Aman et al., 2018*; *Bae et al., 2018*; *Cox et al., 2018*; *Iwasaki et al., 2018*; *Rasmussen et al., 2018*). Given the presence of all steps of scale development in our sampled timepoint, we captured single cells along the continuum of SFC differentiation from pre-SFCs to mature, matrix-secreting cells. Scale development also coincides with expansion of the three major epidermal cell types—periderm (also known as superficial epidermal cells), suprabasal cells, and basal cells—of which basal cells give rise to both of the other cell types (*Lee et al., 2014*). Our transcriptomic analyses confirmed that SFCs express *sp7* and suggested that they likely differentiate from *runx2b+* progenitors (*Figure 2A*). Comparing *runx2b* and *sp7* mRNA distributions during scale development revealed that *runx2b* is consistently more broadly expressed than *sp7*, suggesting the existence of a restricted halo of SFC progenitors surrounding the growing scale (*Figure 2B and D*). To confirm that these cells represent SFC progenitors, we exploited the differential localization of two transgenic reporters, cytosolic ET37:EGFP, labeling all dermal cells but having markedly reduced expression in SFCs, and photoconvertible, nuclear-localizing *sp7*:nEOS, restricted to differentiated SFCs (*Parinov et al., 2004*; *Aman et al., 2021*): if peripheral *runx2+* cells give rise to SFCs, initially cytosolic reporter expression should transition to nuclear reporter expression. We excluded previously differentiated *sp7*:nEOS+ SFCs from consideration by photoconversion (green → red) and looked for ET37:EGFP+ cells (green cytosol) newly expressing *sp7*:nEOS (green nucleus, without red). As expected, over 3 d of scale development, we observed numerous cells acquire nuclear nEOS expression as they were incorporated into growing scales, supporting the inference that peripheral, presumptively *runx2b+* dermal cells differentiate as SFCs (*Figure 2C and D*).

Because the stage at which cells were collected contains SFCs at all steps of differentiation, and because the transcriptomes of SFCs displayed a continuous path between pre-SFC and differentiated SFC in UMAP space (*Figure 2A*), we anticipated that the cell fate transition during SFC differentiation would be captured by pseudotemporal ordering, which can reveal cell state transitions in asynchronous populations of differentiating cells (*Trapnell et al., 2014*). To test this idea, we compared gene expression in UMAP space to spatial patterns of gene expression as revealed by in situ hybridization using a combination of new staining and previously published staining (*Aman et al., 2018*; *Cox et al., 2018*; *Iwasaki et al., 2018*; *Rasmussen et al., 2018*). We predicted that cells having the intermediate state predicted by pseudotemporal ordering should be present circumferentially, between the halo of *runx2+* dermal cells and differentiated *sp7+* SFCs. Instead, these pseudotemporally intermediate cells were located at scale radii, a relatively late-appearing structure comprising cells unlikely to be in a transitional state of SFC differentiation (*Figure 2—figure supplement 1G*). To resolve the true SFC differentiation trajectory, we fate-mapped individual SFCs in vivo by photoconverting *sp7*:nEOS+ SFCs within specific scale regions and following them over several days (*Figure 2E*, *Figure 2—figure supplement 2A*). These analyses showed that SFCs at the posterior margin are progressively displaced towards the scale focus, presumably by addition of newly differentiated SFCs from *sp7*-negative progenitors (*Figure 2E*, top). SFCs in the sub-marginal region contributed to elongated marginal cells,

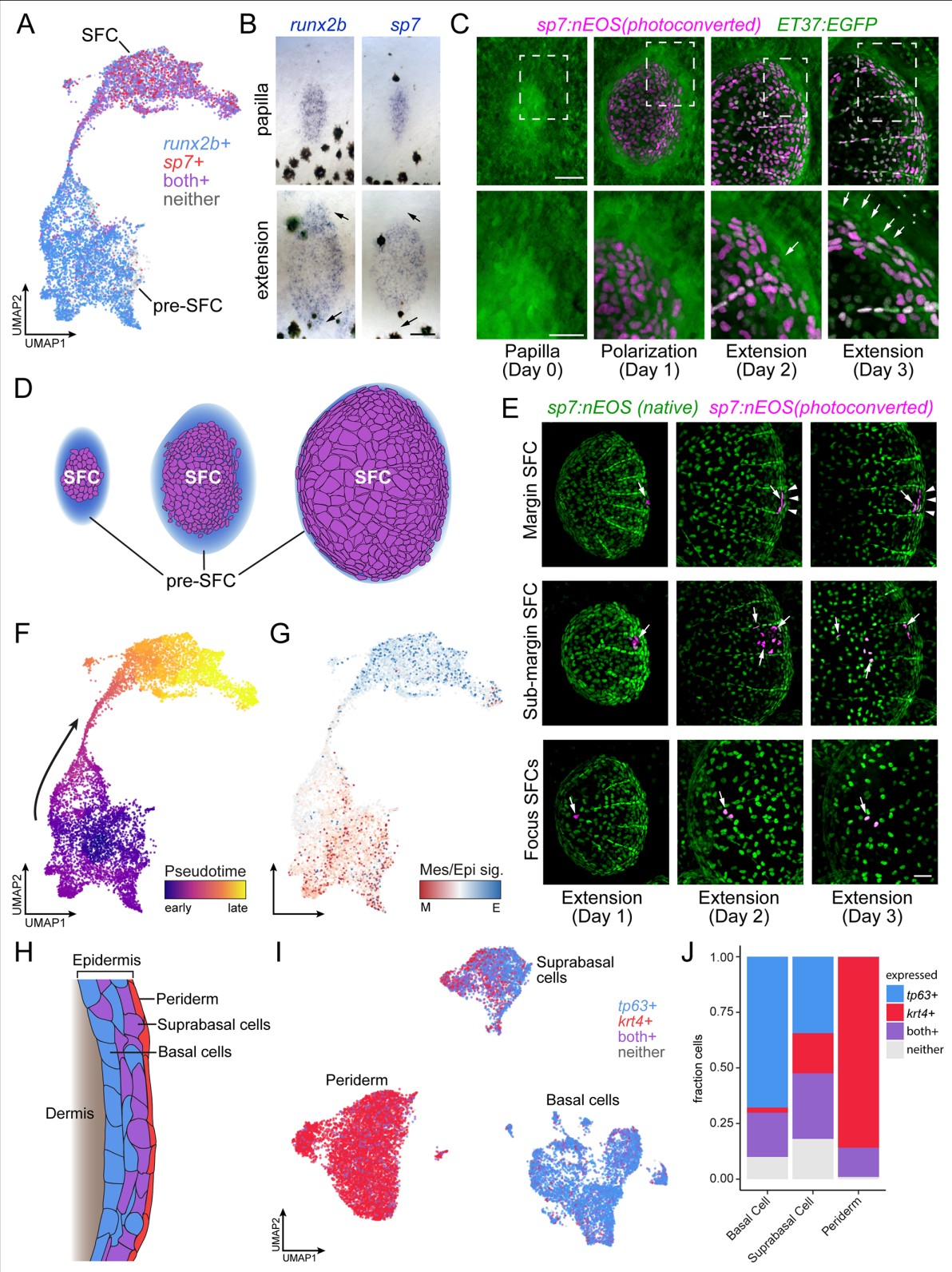

**Figure 2.** Postembryonic skin cell lineage relationships are not reflected in UMAP space. (**A**) UMAP visualization showing distribution of differentiated scale-forming cell (SFC) expressing *sp7* and pre-SFC progenitors expressing *runx2b*. (**B**) In situ hybridization of *sp7* and *runx2b* shows that a halo of pre-SFC progenitors surround the growing scale (arrows). (**C**) *sp7:nEOS*-expressing differentiated SFC (magenta) were labeled by photoconversion on day 1. Over the following 2 d, newly differentiated, un-photoconverted SFC appeared at the scale margin (arrows; n = 5 fish). (**D**) Schematic representation of

*Figure 2 continued on next page*

*Figure 2 continued*

differentiated SFC (purple) and the associated halo of pre-SFC (blue). (**E**) Photoconversion of small groups of SFC in the scale margin and sub-margin; and single-cell photoconversion of focus SFCs (arrows) showed that SFC are progressively displaced toward the scale focus and that SFC in all these regions are capable of cell division (arrows, n ≥ 4 fish for each region tested). Margin SFCs were displaced toward the posterior by newly differentiated, un-photoconverted SFCs (arrowheads). (**F**) SFCs in UMAP space colored by 'pseudotime' rooted in the SFCs. (**G**) SFCs in UMAP space colored by the ratio of a mesenchymal (migratory) signature to an epithelial signature (***Supplementary file 2***—Table 3). (**H**) Schematic representation of epidermis with major substrata. (**I**) UMAP visualization of wild-type epidermis, subclustered independently of other cell types and displaying expression of the epidermal basal cell marker *tp63* (blue) and the periderm marker *krt4* (red). (**J**) The fraction of cells from panel (**H**) that pass a minimum threshold for expression of *tp63*, *krt4*, or both genes. Scale bars, 50 μm (**B, C, E**); 25 μm, (**C**, lower).

The online version of this article includes the following figure supplement(s) for figure 2:

**Figure supplement 1.** In situ hybridization using probes against specific transcripts localizes heterogeneous scale-forming cell (SFC) in the developing scale.

**Figure supplement 2.** Scale-forming cell (SFC) lineage and proliferation.

scale radii cells, and an SFC subset that rapidly displaced toward the scale focus (***Figure 2E***, middle). Lastly, cells at the focus in a nascent scale remained in the focus as the scale grew (***Figure 2E***, bottom). These results demonstrate that radii cells descend from marginal SFCs, consistent with previous live imaging results (***Cox et al., 2018***; ***Iwasaki et al., 2018***; ***Rasmussen et al., 2018***), confirming that SFCs do not pass through an intermediate state as scale radii cells during differentiation, which a facile interpretation of pseudotemporal ordering might suggest. Thus, although cell state transitions inferred from transcriptomes alone can suggest cell lineage relationships and cell states along a differentiation continuum, we find that the pseudotemporal ordering, while continuous from pre-SFC to differentiated SFC, did not faithfully represent the cell state path during SFC differentiation. Instead, the continuity of cell states revealed by pseudotemporal ordering could reflect continuous variation in other biological processes apart from differentiation, like states of the cell cycle or migration (***Haghverdi et al., 2015***; ***Saunders et al., 2019***). Given the mesenchymal appearance of pre-SFC dermal cells and the epithelial appearance of differentiated SFCs (***Figure 2C***; ***Figure 1—figure supplement 1***) (***Iwasaki et al., 2018***; ***Aman et al., 2021***), we predicted that pseudotemporal order in this dataset might reflect differences in gene expression associated with these different cellular morphologies. To test this possibility, we constructed gene expression signatures for mesenchymal and epithelial states from the literature and mapped the ratio of epithelial-to-mesenchymal scores on cells in UMAP space, which revealed an overall correspondence of these scores with pseudotemporal order (***Figure 2F and G***). Finally, our observations also provided an opportunity to resolve a controversy as to whether differentiated SFCs are capable of proliferating or whether scale growth occurs exclusively by hypertrophic growth of individual cells (***Cox et al., 2018***; ***Iwasaki et al., 2018***): both transcriptomic analyses and live imaging experiments showed that differentiated SFCs remain proliferative (***Figure 2—figure supplement 2B–D***).

In addition to SFC differentiation, we sought to understand the transitional dynamics of epidermal differentiation during post-embryonic skin maturation. During this stage of development, basal epidermal cells are the stem cell population that differentiate into both suprabasal and periderm cells, and each of the three major epidermal cell types are well represented in our dataset (***Figure 2H and I***, ***Figure 1—figure supplement 3***; ***Guzman et al., 2013***; ***Lee et al., 2014***). While periderm cells at the sampled stage are likely of dual origin, representing a mixture of early embryonic and stem cell-derived cells, suprabasal cells are entirely derived from basal cells (***Kimmel et al., 1990***; ***Guzman et al., 2013***; ***Lee et al., 2014***). Given the known lineage relationships, we predicted that the single-cell transcriptome data would reveal a continuum of cell state transitions during the differentiation of both cell types. While the established cell type markers for basal cells and periderm cells, *tp63* and *krt4*, displayed a clear transition between basal cells (*tp63+*, *krt4-*), suprabasal cells (*tp63+*, *krt4+*), and periderm (*tp63-*, *krt4+*), the trajectories were discontinuous in UMAP space (***Figure 2I and J***). These discontinuities were present in both global and tissue-specific UMAP projections and across a wide range of UMAP parameters (***Figure 2I***, ***Figure 1—figure supplement 3***). Moreover, the number of differentially expressed genes between each of the epidermal types was about twice as large as between cell clusters having continuous trajectories (basal cell vs. periderm, 7341 DEGs; basal cell vs. suprabasal, 4024 DEGs; pre-SFC vs. SFC, 2373 DEGs; all q < 0.01), suggesting that discontinuities among epidermal cell subtypes reflect abrupt transcriptional changes during differentiation, rather

than artifacts of the dimensionality reduction itself. Together, our results highlight the importance of coupling true lineage information with well sampled, high-resolution single-cell transcriptomes in order to understand complex transcriptional dynamics over the course of differentiation in vivo.

## A heterogeneous population of epidermal and dermal cells contributes to scale plate formation

Zebrafish elasmoid scales represent one of several forms of calcified skin appendages that cover the bodies of most non-tetrapod fish species (hereafter referred to as 'fish'). Calcified skin appendages are an ancient vertebrate trait with a robust fossil record, first appearing in the Ordovician 450 million years ago, prior to the appearance of paired fins, jaws, and teeth (*Smith et al., 2002*; *Märss, 2006*; *Sire et al., 2009*; *Fraser et al., 2010*; *Turner et al., 2010*; *Janvier, 2015*). These ancestral skin appendages were morphologically and compositionally similar to modern-day teeth, with a hypercalcified, enamel-like matrix capping collagen-rich calcified matrices resembling dentin, bone, or both (*Quan et al., 2020*; *Smith et al., 2002*; *Märss, 2006*; *Sire et al., 2009*; *Figure 3A*). While certain extant non-teleost fishes, like sharks, bichir, and gar, have scales that resemble skin appendages of ancient fishes and modern teeth, the flattened morphology and elastic flexibility of elasmoid scales typical of most extant fish are highly derived (*Quan et al., 2020*; *Sire et al., 2009*). Histological and ultrastructural studies have shown that the elasmoid scales of zebrafish and other teleosts are composed of weakly calcified collagenous matrix, known as elasmoidin, capped by a collagen-free, hypermineralized limiting layer that forms in close proximity to basal epidermal cells with an overtly secretory morphology (*Figure 3A*; *Quan et al., 2020*; *Sire et al., 1997b*; *Quan et al., 2020*). In vertebrate teeth and tooth-like scales of non-teleost fish, layers of calcified matrix are deposited by two cooperating cell types, mesenchymal cells of dermal or neural crest origin that form collagen-rich calcified matrix like dentin, bone or both, and overlying epithelial cells of epidermal or endodermal origin that produce hypermineralized matrices like enamel (*Quan et al., 2020*; *Sire and Huysseune, 2003*; *Oralová et al., 2020*; *Kawasaki et al., 2021*).

Hypothesizing that epidermis of zebrafish retains this ancient enamel deposition function, we predicted that a subset of epidermal basal cells would express transcripts encoding enamel matrix proteins (EMPs). EMPs were originally discovered in human patients suffering from amelogenesis imperfecta, a congenital condition characterized by defective enamel formation (*Smith et al., 2017*). EMPs are part of family of proteins known as secretory calcium-binding phosphoproteins (SCPPs) that are critical for calcification of bone, dentin, and enamel (*Kawasaki, 2009*). The zebrafish genome harbors two previously identified EMP-gene orthologs, including orthologs of human Enamelin (encoded by enam) and Ameloblastin (encoded by ambn), in addition to fish-specific orthologs that likely originated by tandem duplication (*Qu et al., 2015*; *Braasch et al., 2016*; *Kawasaki et al., 2017*). We therefore analyzed the distribution of SCPP-encoding transcripts, predicting that EMP-transcript-expressing cells would be found among basal epidermal cells. Consistent with our prediction, we identified a transcriptionally distinct population of basal epidermal cells, separated from other basal epidermal cells in UMAP projections, that expressed EMP genes (*Figure 3B and C*). Strikingly, transcripts of the conserved EMP gene ambn were found almost exclusively in this subpopulation of basal cells, which we refer to as EMP+ epidermal cells (*Figure 3B and C*). A previous ultrastructure study revealed epidermal basal cells with a secretory morphology in contact with the calcified scale matrix (*Sire et al., 1997b*). To test correspondence of that population with EMP+ epidermal cells identified in our transcriptomic data, we visualized ambn expression by in situ mRNA hybridization, which revealed that EMP+ epidermal cells are positioned precisely where epidermal cells contact the scale matrix (*Figure 3D–F*).

Although enamel-like capping matrix is common in phylogenetically diverse vertebrates (*Figure 3A*; *Sire et al., 2009*), it is possible that epidermal expression of EMP transcripts evolved convergently in zebrafish. If epidermal EMP+ basal cells are homologous with mammalian ameloblasts, we reasoned that in addition to genes encoding matrix proteins, these cells should express transcription factors that regulate mammalian ameloblast differentiation. Indeed, we find that *dlx3a*, *dlx4a*, *msx2a*, and *runx2b* are expressed in these cells (*Figure 3B*; *Pemberton et al., 2007*; *Debiais-Thibaud et al., 2011*; *Urzúa et al., 2011*; *Bae et al., 2018*; *Chu et al., 2018*; *Woodruff et al., 2022*; *Li et al., 2011b*). Together, these results suggest that the enamel-like, hypermineralized limiting layer of the scale is produced by deeply conserved, ameloblast-like epidermal cells.

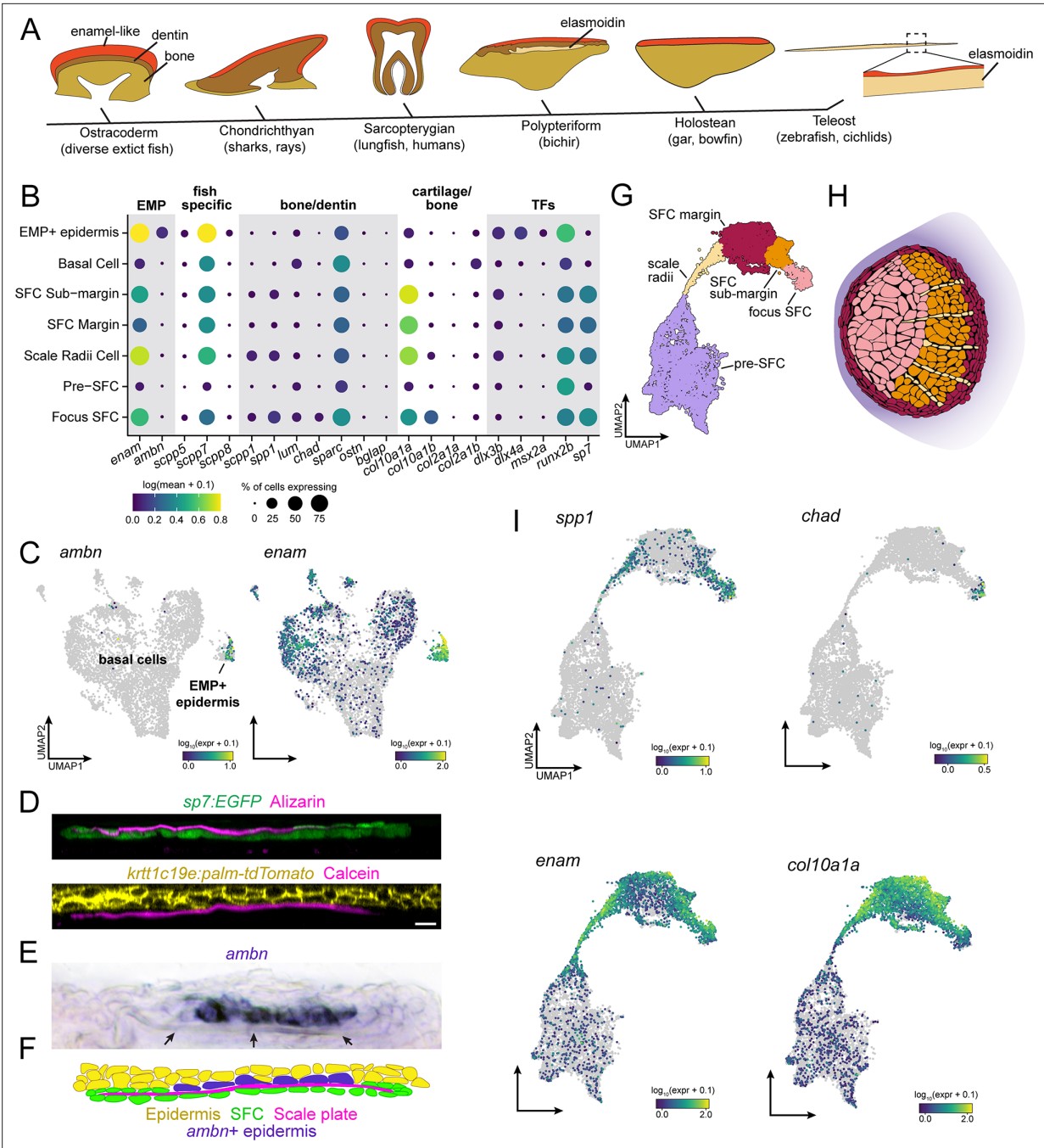

**Figure 3.** Evidence for epidermal and dermal contributions to scale plate ECM. (**A**) Simplified vertebrate phylogeny with schematic depictions of calcified appendages. Phylogeny based on *Near et al., 2012*; *Betancur-R et al., 2013*; schematics drawn after (*Sire et al., 2009*). (**B**) Dotplot visualization of transcripts encoding enamel matrix proteins (EMPs), fish-specific secretory calcium-binding phosphoprotein (SCPP) tandem duplicates, non-collagen matrix proteins associated with bone and dentin, collagen associated with cartilage and transcription factors that regulate osteoblast and ameloblast differentiation. (**C**) UMAP visualization of epidermal basal cells showing distribution of transcripts encoding EMPs. (**D**) Optical sections of growing scale in live animals showing the relative position of calcified matrix dyed with Alizarin Red S (ARS) or Calcein, and dermal scale-forming cell (SFC) visualized with *sp7:EGFP* transgene and epidermis visualized with *krtt1c19e:palm-tgTomato* transgene. (**E**) In situ hybridization of *ambn*, encoding the EMP Ameloblastin. Arrows point to the calcified scale plate. (**F**) Schematic representation of epidermal *ambn* expressing cells (blue), the calcified scale plate (magenta), dermal SFC (green), and epidermis (yellow). (**G**) UMAP visualization of dermal SFC and pre-SFC. (**H**) Position of SFC sub-types within a scale inferred from in situ hybridization assays (*Figure 2—figure supplement 1*). (**I**) UMAP visualization of transcripts encoding non-collagen matrix proteins associated with bone (*spp1, chad*), enamel (*enam*), and cartilage (*col10a1a*). Scale bar, 10 μm.

In teeth and tooth-like skin appendages, enamel-like matrix caps more collagen-rich calcified matrices such as dentin or bone that are deposited by condensed mesenchymal cells (*Sire et al., 2009*; *Fraser et al., 2010*; *Martens et al., 2021*; *Arola et al., 2018*). Although dermal SFCs are frequently referred to as 'osteoblasts' due to their association with calcified matrices and their expression of conserved transcription factor genes *sp7* and *runx2a/b*, the elasmoidin matrix they deposit, characterized by weakly calcified, plywood-like layers of hydrated collagen fibrils, is materially distinct from bone or dentin (*Sire et al., 2009*; *Metz et al., 2012*). We therefore hypothesized that SFCs would express a distinct complement of SCPP transcripts. To elucidate this repertoire and compare dermal SFCs with osteogenic cell types like osteoblasts, odontoblasts, and ameloblasts, we assessed the expression SCPP transcripts. For these analyses, we subclustered the SFCs/pre-SFCs and identified five major cell states based on gene expression from sci-RNA-seq and in situ hybridization (*Figure 3G and H*, *Figure 2—figure supplement 1*). Consistent with an overall similarity between osteoblasts and SFCs, we detected transcripts of Osteopontin (*spp1*) (*Figure 3B and I*) and Chondroadherin (*chad*) (*Figure 3B and I*, *Figure 2—figure supplement 1J*), encoding bone matrix proteins, in addition to Secretory-calcium-binding Phosphoprotein 1 (scpp1), which is homologous to human Dentin-Matrix-Acidic Proteins that form a component of both bone and dentin (*Figure 3B and I*; *Larsson et al., 1991*; *Raouf et al., 2002*; *Hessle et al., 2013*; *Venkatesh et al., 2014*; *Braasch et al., 2016*; *Kawasaki et al., 2017*). Nevertheless, we failed to detect robust expression of transcripts encoding bone matrix proteins Osteocrin (*ostn*) or Osteocalcin (bone gamma-carboxyglutamate protein [bglap]) in SFCs (*Figure 3B*; *Thomas et al., 2003*). In addition to a subset of bone-specific transcripts, SFCs also expressed EMP genes and genes encoding fish-specific SCPPs (*Figure 3B and I*). Expression was spatially restricted among SFCs, with the bone-associated transcripts, *spp1* and *chad*, primarily expressed in the scale focus and radii SFCs, while EMP genes were restricted to SFCs at the scale margin (*Figure 3B, G, H and I*, *Figure 2—figure supplement 1J*).

Scale elasmoidin is a flexible, collagenous ECM (extracellular matrix), material properties that are similar to cartilage (*Quan et al., 2020*). We therefore wondered whether dermal SFCs express matrix proteins associated with cartilage formation. *Col10a1* and Col2a1 are major structural molecules in collagen, although *col10a1* transcription has also been documented in osteoblasts (*Gu et al., 2014*; *Yang et al., 2014*; *Kawasaki et al., 2021*). The zebrafish genome harbors genes encoding two *Col10a1* orthologs (*col10a1a* and *col10a1b*), and we found both transcripts in SFCs representing distinct steps of maturation; however, transcripts of Col2a1 genes (col2a1a and col2a1b) were not robustly detected in these cells (*Figure 3B and I*, *Figure 2—figure supplement 1F and I*). Transcripts encoding additional factors associated with mineralized matrix formation, such as Osteonectin (sparc), were expressed broadly in skin (*Figure 2A and H*), raising the possibility of additional roles beyond assembly of calcified matrix (*Rosset and Bradshaw, 2016*).

Together, these analyses revealed the presence of epidermal EMP expressing cells and support a hypothesis of ancient homology between ameloblast-like cells in fish skin and the mammalian dental lamina, and between the enamel-like materials that coat zebrafish scales and tetrapod teeth. Furthermore, our observations that dermal SFCs express a subset of genes associated with bone development, as well as genes encoding EMPs and cartilage proteins, suggest that the distinct properties of elasmoid matrix are due to a distinct complement of matrix proteins. While these conclusions are correlational, this work will facilitate functionally testing the role of candidate matrix proteins in the material properties of calcified matrices.

## Eda-Edar-NF-κB and TH regulate distinct stages of dermal SFC development

Eda-Edar-NF-κB signaling plays conserved roles in regulating the patterning and morphogenesis of vertebrate skin appendages (*Cui and Schlessinger, 2006*; *Lefebvre and Mikkola, 2014*), whereas TH regulates multiple aspects of skin development and homeostasis (*Mancino et al., 2021*). Both *eda* mutants and hypoTH fish completely lack scales at the stage sampled, allowing us to compare transcriptomic signatures and cell type complements associated with scale loss in these different backgrounds (*Figure 1A*). Analyses of cell type abundance revealed that both scale-free conditions were characterized by a complete lack of fully differentiated dermal SFCs (*Figure 4A and B*). These analyses further showed that eda mutants—despite homozygosity for a presumptive null allele—retained a small subset of dermal pre-SFC progenitors, whereas hypoTH skins lacked pre-SFC entirely. We

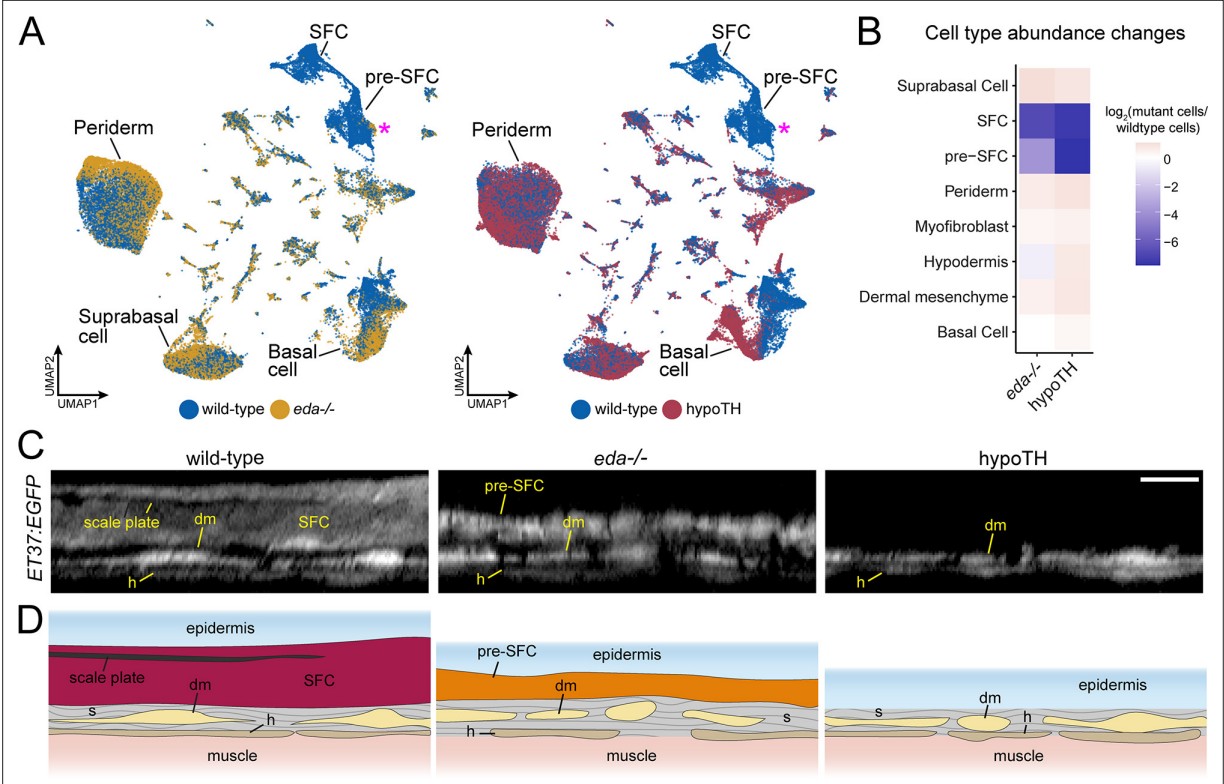

**Figure 4.** eda and thyroid hormone (TH) regulate signaling ligand transcription in basal epidermal cells. (**A**) UMAP visualizations with wild-type cells in blue, *eda* mutant cells in yellow, and hypoTH cells in red highlighting the absence of scale-forming cell (SFC) and residual pre-SFC in skins of *eda* mutants and the absence of both populations in skins of hypoTH fish. Magenta asterisks mark differences in pre-SFC complements between *eda* mutant and hypoTH; other cell types designated in *Figure 1C*. (**B**) Heatmap visualization of cell type abundance shows that *eda* mutants retain more pre-SFCs than hypoTH fish. (**C**) Optical sections and (**D**) schematic representations of wild-type, *eda* mutant and hypoTH skin, visualized in *ET37:EGFP* transgenics, showing abundance of SFC in wild-type, a thin layer of pre-SFC in *eda* mutants, and lack of pre-SFC in hypoTH skin. dm, dermal mesenchyme; h, hypodermis; s, stromal collagen. Scale bar, 10 μm (**C**).

have previously shown that hypoTH fish lack superficial dermal cell types at the stage sampled for sequencing, though these cell types do appear later in development (*Aman et al., 2021*). To confirm the presence of residual, pre-SFC in eda mutants, we imaged live fish expressing ET37:EGFP (*Parinov et al., 2004*; *Aman et al., 2021*). Indeed, eda mutants exhibited a population of ET37:EGFP+ dermal cells—presumptive pre-SFC—beneath the epidermis that was not present in hypoTH fish (*Figure 4C and D*).

In previous work, we found that Eda-Edar-NF-κB signals are transmitted from the dermis to epidermis during scale morphogenesis (*Aman et al., 2018*). The apparently monogamous receptor for Eda, encoded by edar, is expressed in the epidermis during scale morphogenesis. The signaling ligand, encoded by eda, has a dynamic expression pattern that shifts from broad expression in unspecified epidermis to localized expression in dermal papillae during scale morphogenesis. These expression dynamics are strikingly similar during mouse hair follicle and chicken feather patterning (*Montonen et al., 1998*; *Harris et al., 2008*; *Houghton et al., 2005*). We confirmed that these expression domains were reflected in our sci-RNA-seq data, in which edar is detected primarily in basal epidermal cells, with most intense expression in the shha+ epidermal placode (*Figure 5A*; *Aman et al., 2018*). eda transcripts were detected at much lower levels and were much more dispersed across cell types than edar transcripts but were nevertheless enriched in pre-SFC as expected (*Figure 5A*).

Since Eda-Edar-NF-κB pathway activation occurs exclusively in epidermis but drives morphogenesis of dermal cells, we predicted the existence of an Eda-dependent signaling ligand expressed in basal epidermal cells that would complete a presumptive epithelial–mesenchymal signaling loop (*Aman et al., 2018*). We previously observed that global misexpression of an Fgf ligand rescued scale development in eda mutant fish, and that Edar expression in epidermal cells, but not dermal cells,

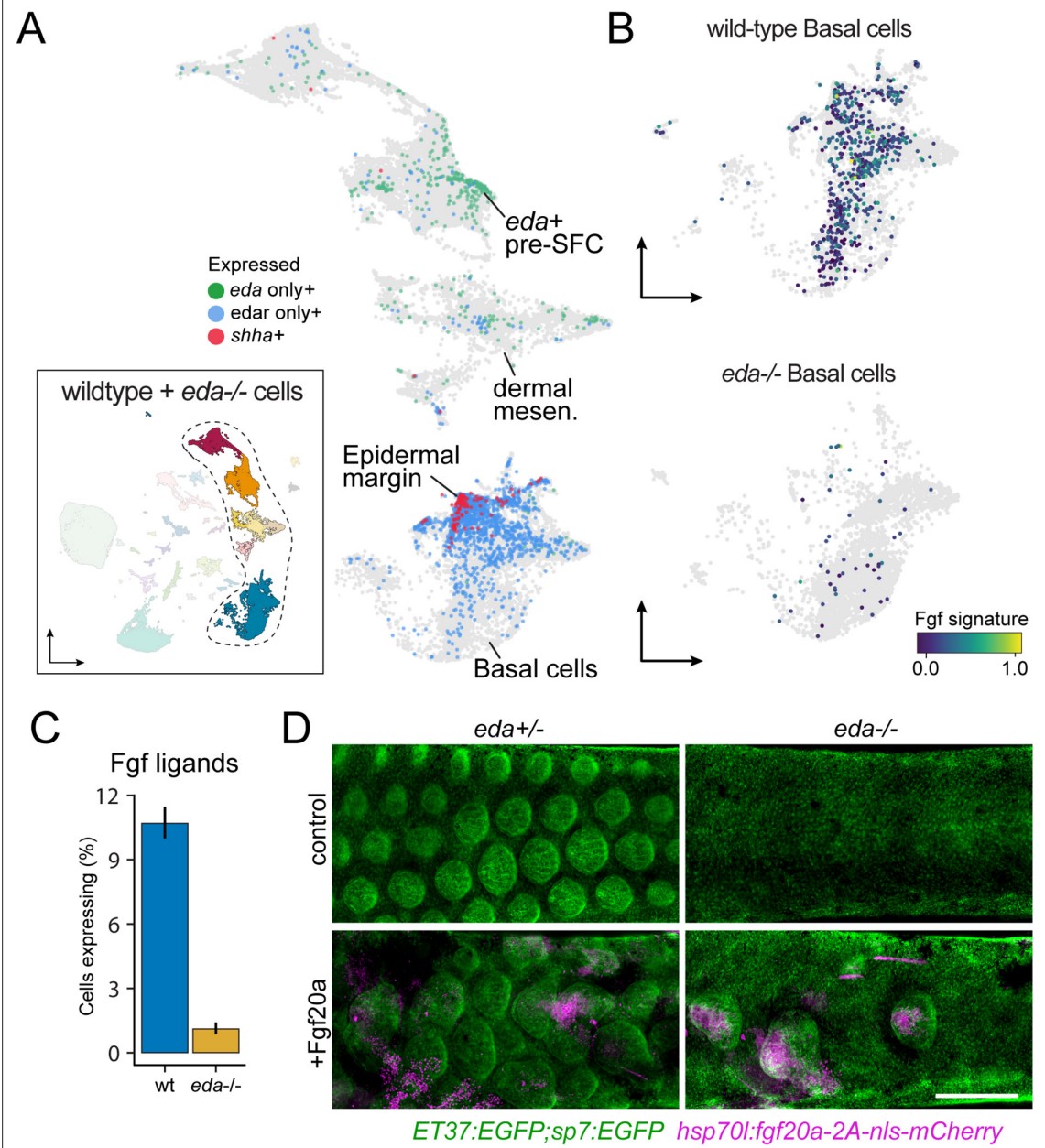

**ET37:EGFP;sp7:EGFP** **hsp70l:fgf20a-2A-nls-mCherry**

**Figure 5.** eda regulates scale-forming cell (SFC) differentiation via transcriptional regulation of basal epidermal Fgf ligands. (**A**) Wild-type dermal and basal epidermal cells plotted in UMAP space and colored by whether they are expressing *eda*, *edar*, or *shha* (*shha+* cells also include cells expressing both *edar* and *shha*). (**B**) Wild-type and Eda mutant basal cells plotted in UMAP space and colored by the expression of the signature score of Fgf ligands that are specifically expressed in basal cells (*fgf24, fgf20a, fgf20b*; specificity score >0.1). (**C**) Percent of cells expressing the Fgf ligand signature between wild-type and Eda mutant basal cells (error bars estimated via bootstrapping [n = 100]). (**D**) Scales and dermis visualized in *sp7:EGFP; ET37:EGFP* double transgenics (green). Heat-shocked control *eda⁺/⁻* larvae developed well-patterned, uniformly shaped scales (n = 6), whereas heat-shocked control *eda⁻/⁻* larvae developed no scales (n = 6). Mosaic heat-shock induction of Fgf20a, stringently selected for expression in epidermal cells (magenta) caused mis-patterned and dysmorphic scales to grow in wild-type *eda⁺/⁻* larvae (n = 25 epidermal clones in six fish) and rescued scale formation in *eda⁻/⁻* larvae (n = 36 epidermal clones in six fish). Scale bar, 500 µm (**D**).

was sufficient to drive scale formation (***Aman et al., 2018***). Those results suggested that Eda signaling drives SFC differentiation via epidermal expression of one or more Fgf ligands. Our transcriptomic analysis showed that, indeed, epidermal expression of Fgf ligands is Eda-dependent (***Figure 5B and C***). Among Fgf ligand transcripts detected in epidermis was that of fgf20a, which plays a conserved role in regulating dermal morphogenesis in amniote skin appendage development and has been

implicated in scale development and regeneration in zebrafish (*Huh et al., 2013*; *Daane et al., 2016*). If Fgf20a functions downstream of epidermal Eda-Edar-NF-κB signaling, we predicted that experimental restoration of fgf20a expression in eda mutant skin should bypass the requirement for Eda in scale formation, thereby rescuing scales even in the absence of Eda function. Indeed, heatshock-driven expression in F0 mosaics stringently selected for basal epidermal expression of Fgf20a in the skin of Eda mutants led to localized rescue of scales where transgene expression was detectable (*Figure 5D*). Notably, fgf20a mutant zebrafish do not have a squamation phenotype unless present in an fgfr1a mutant background, suggesting functional redundancy among Fgf ligands (*Daane et al., 2016*). Those results and our present findings together suggest that Eda-Edar-NF-κB signaling regulates SFC differentiation via multiple epidermal Fgf ligands, including Fgf20a.

Unlike Eda signaling, very little is known of potential TH targets in the skin. Therefore, we compared between wild-type and hypoTH backgrounds the expression of transcripts within each of the five major cell types. This analysis revealed substantial differences in gene expression between backgrounds in dermal and epidermal cell types, and particularly in basal cells of the epidermis (*Figure 6A*). Given the absence of superficial pre-SFC in hypoTH skin (*Figure 4B–D*), we hypothesized that TH regulates the expression of cues in epidermal basal cells that recruit dermal cells to the most superficial layer of the dermis, subjacent to the epidermis, prior to scale papilla formation (*Aman et al., 2021*). To test this idea, we examined ligand genes with detectable expression across each of the seven major pathways between wild-type and hypoTH basal cells, which revealed markedly reduced expression for several non-FGF MAPK ligand genes, including PDGFα orthologs (pdgfaa, pdgfab) that are known in amniotes to regulate mesenchymal cell motility and proliferation (*Figure 6B–D*; *Karlsson et al., 1999*). If PDGFα ligands are responsible for recruiting dermal cells similarly in zebrafish skin, then restoring expression of pdgfaa in basal cells of the epidermis in hypoTH fish should rescue the formation of superficial dermal cells in this background. When we forced expression of Pdgfaa in basal cells of epidermis by heatshock induction and stringent selection of basal epidermal expression in F0 mosaics, we found, as predicted, a recruitment of dermal cells in hypoTH skin, leading to a locally stratified dermis (*Figure 6E*) similar to that of the wild-type (*Figure 4C*). Early, heatshock-induced Pdgfaa expression also led to precocious dermal stratification in wild-type and eda mutant fish (*Figure 6—figure supplement 1A*). Pdgfaa expression did not, however, rescue the onset of squamation in hypoTH fish, which begins at a larger size and only after about twice the time as in wild-type fish (14 SSL vs. 9 SSL, 40 vs. 21 d postfertilization in our rearing conditions) (*Aman et al., 2021*). Nor did Pdgfaa lead to mis-patterned and dysmorphic scales in wild-type fish, or a rescue of squamation in eda mutants, as we observed for Fgf20a (*Figure 5D*). Together, these observations suggest that Pdgfaa-dependent stratification of dermis and Fgf-dependent differentiation of SFC are functionally decoupled processes that occur sequentially during skin morphogenesis (*Figure 6—figure supplement 1B*). Because Pdgfaa rescued dermis stratification, but not scale development in hypoTH skin, we predict that additional TH transcriptional targets regulate skin morphogenesis. Indeed, differential expression analyses suggested several excellent candidates for mediating additional signals from epidermal basal cells to pre-SFC (*Figure 6—figure supplement 1C*).

These observations from scaleless skins indicate that epidermal basal cells are critical targets for TH and Eda-Edar-NF-κB signals, and that epidermally expressed Pdgfaa and Fgf ligands link TH and Eda signaling to dermal cell recruitment and dermal papilla development, respectively (*Figure 7*). These findings are of potential clinical significance as the pathophysiologies underlying TH skin diseases remain unclear (*Mancino et al., 2021*).

## Hypodermal contribution to the microenvironment of stripe-forming pigment cells

Zebrafish pigment patterning is a useful study system for elucidating the principles that govern developmental patterning and post-embryonic developmental progression (*Patterson and Parichy, 2019*). The alternating dark stripes of melanophores with sparse blue-tinted iridophores and light interstripes of yellow-tinted iridophores with orange xanthophores form deep in the skin (*Hessle et al., 2013*; *Guzman et al., 2013*), which remains remarkably transparent throughout the life of the animal (*Figure 1A and B*). Although pigment cells are an integral part of the skin and comprise its major visual element, we know little about how these cells interact with other cell types in this microenvironment.

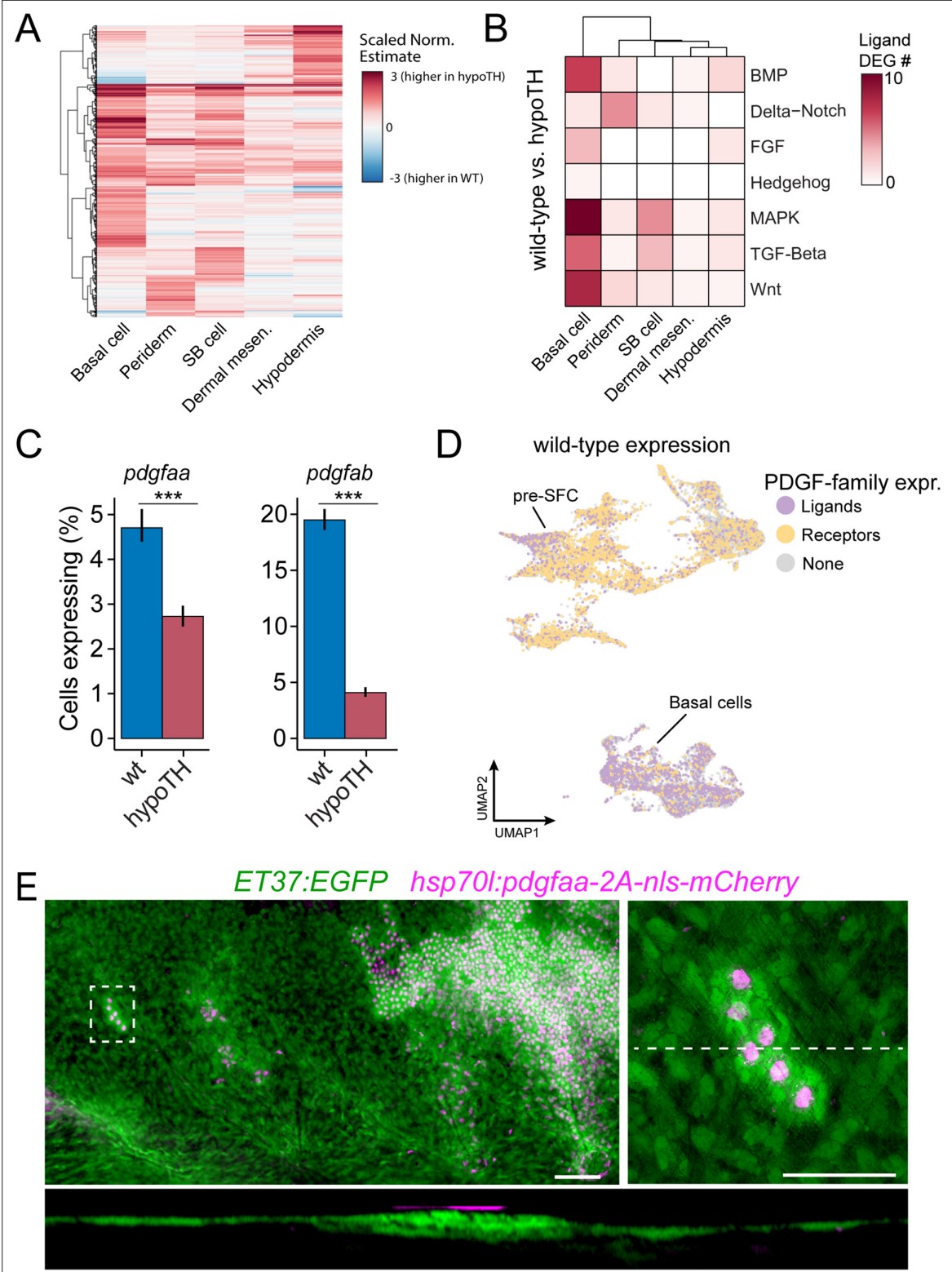

**Figure 6.** Thyroid hormone (TH) drives dermal stratification via transcriptional regulation of epidermal Pdgfα expression. (**A**) Differential gene expression analysis between cell types of wild-type and hypoTH fish revealed extensive changes in expression across dermal and epidermal cell types (n = 836 genes, q-value < 0.01, normalized effect >2) (***Supplementary file 2***—Table 4). SB cell, suprabasal cell. (**B**) Of the differentially expressed genes, ligands of major signaling pathways involved in morphogenesis are also enriched in basal cells. (**C**) Both *pdgfaa* and *pdgfab* ligands are differentially expressed (***q-value < 1e-10) between wild-type and hypoTH basal cells of the epidermis (error bars estimated via bootstrapping [n = 100]). (**D**) Wild-

*Figure 6 continued on next page*

Figure 6 continued

type dermal and basal cells of epidermis plotted in UMAP space and colored by whether they express *pdgfaa*, *pdgfab,* or both (ligands) as well as *pdgfra*, *pdgfrb,* or both (receptors). (**E**) Upper left: mosaic heat-shock induction of Pdgfaa (magenta), stringently selected for expression in epidermal basal cells, rescued stratification of hypoTH dermis, visualized with *ET37:EGFP* (green) (n = 65 clones in eight fish). Upper right: higher magnification of boxed area showing accumulation of dermal cells underneath Pdgfaa+ epidermal cells. Bottom: optical cross section of boxed area reveals multiple dermal layers only in proximity to Pdgfaa+ epidermal cells. Scale bars, 50 µm (**E**, upper-left panel), 10 µm (**E**, enlarged region, upper-right and lower panels).

The online version of this article includes the following figure supplement(s) for figure 6:

**Figure supplement 1.** Excess Pdgfaa expression led to precocious dermal stratification in wild-type and *eda* mutant fish.

To better define how pigment cells are integrated with other skin cell types and identify tissue environmental factors that may influence pigment patterning, we included in our study the bonaparte mutant, homozygous for a presumptive loss of function mutation in basonuclin 2 (bnc2) (*Lang et al., 2009*). bnc2 mutants have a very sparse complement of pigment cells as adults owing to progressive pigment cell death during the larva-to-adult transition, with iridophores especially affected (*Lang et al., 2009*; *Patterson and Parichy, 2013*).

At the stage of tissue collection, fewer iridophores and melanophores were evident in bnc2 mutants compared to wild-type controls, mirroring prior quantitative comparisons (*Figure 8A*). Previous analysis of genetic mosaics demonstrated that Bnc2 function is required in the hypodermis for survival and patterning of pigment cells (*Lang et al., 2009*). Accordingly, we predicted that wild-type and bnc2 mutant fish should have substantial differences in gene expression within the hypodermal cell population. Yet alignment of UMAP projections for wild-type and bnc2 mutant cells revealed instead a profound deficiency of hypodermal cells themselves (*Figure 8B*). We quantified the proportion of

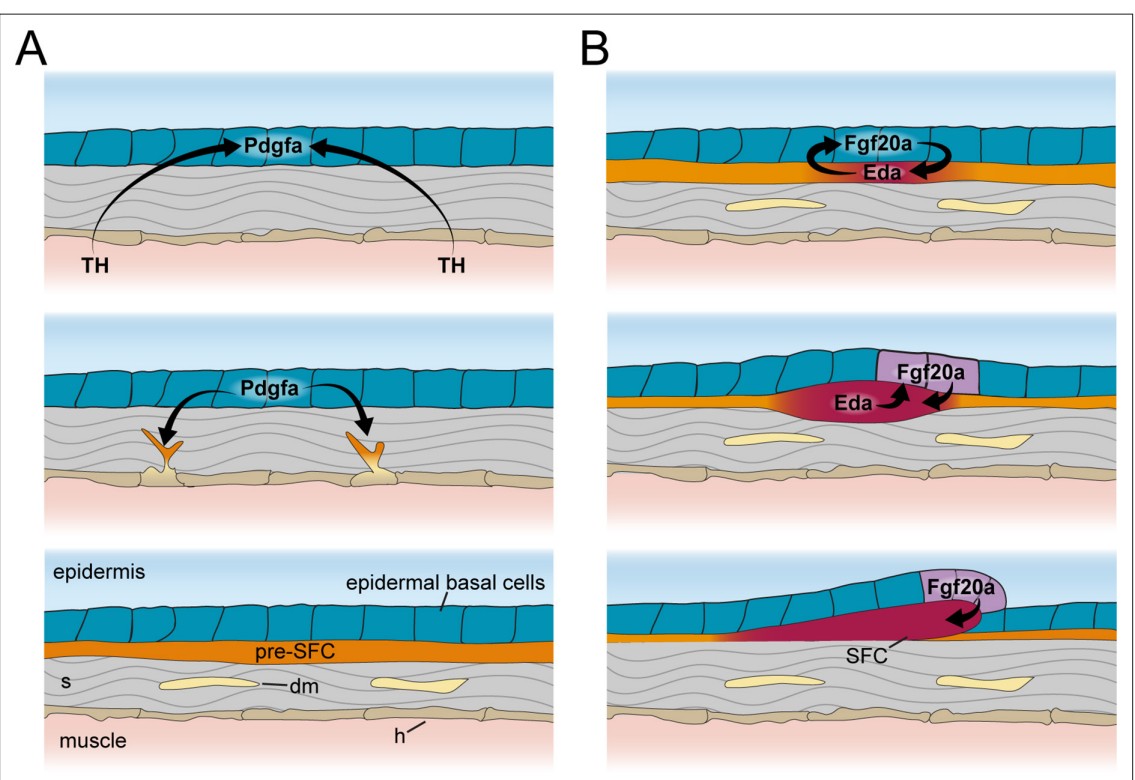

**Figure 7.** Schematic representation of signaling interactions that regulate dermal morphogenesis. (**A**) Globally circulating thyroid hormone (TH) stimulates expression of transcripts encoding Pdgfα ligand in basal epidermal cells (blue). Epidermally expressed Pdgfa ligand regulates migration and differentiation of dermal mesenchyme in the stroma and pre-scale-forming cells (pre-SFCs) that accumulate just beneath the epidermis prior to scale development. (**B**) Epidermally expressed Fgf20a ligand stimulates differentiation of SFCs in the superficial pre-SFC population. Subsequently, SFCs express Eda ligand, which, in turn, maintains expression of Fgf20a. The mechanisms that initiate squamation and regulate hexagonal scale patterning remain unknown.

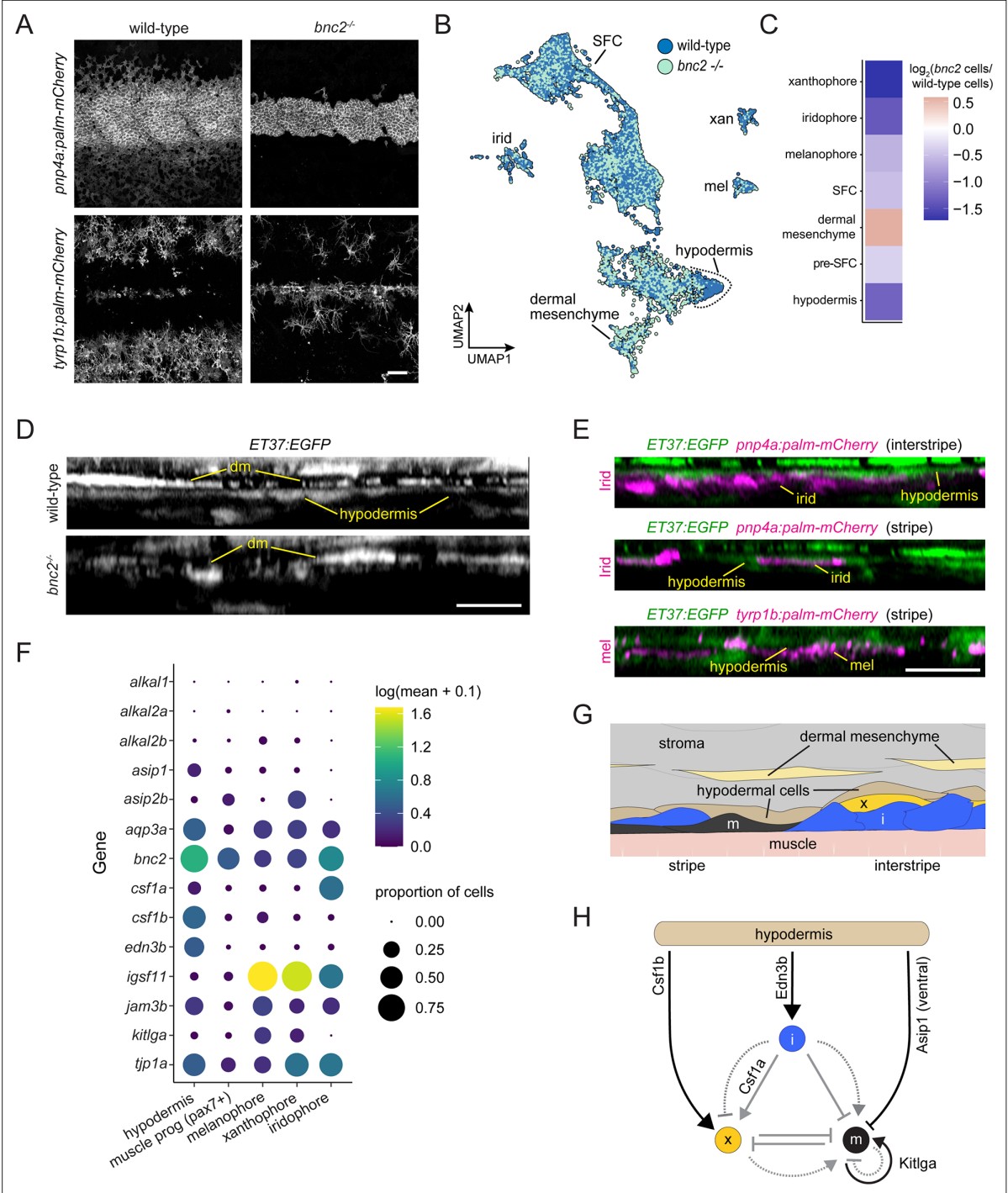

**Figure 8.** Hypodermis is a pigment cell support cell population. (**A**) At 9.6 standardized standard length (SSL), *bonaparte* mutants have grossly fewer iridophores (*pnp4a:palm-mCherry*) and melanophores (*tyrp1b:palm-mCherry*). (**B**) UMAP visualization of dermal cells and pigment cells with wt in blue and *bnc2* mutant in red shows specific deficiency in hypodermal cells. (**C**) Heatmap showing the log$_2$ proportion of dermal cell subtypes and pigment cells. (**D**) Orthogonal projections of live, super-resolution imaging of dermal cells in wild-type and *bnc2* mutants expressing *ET37:EGFP*. Wild-type hypodermis is a thin, confluent cell layer underneath the more brightly labeled dermal mesenchyme (dm). Stage-matched *bnc2* mutant dermis had dermal mesenchyme, but lacked a hypodermal layer. (**E**) Live imaging of fish doubly transgenic for *ET37:EGFP* to visualize hypodermis, and *pnp4a:palm-mcherry* to visualize iridophores (irid) or *tyrp1b:palm-mCherry* to visualize melanophores (mel). Both pigment cell types reside in close contact with hypodermal cells. (**F**) Dotplot heatmap showing expression level of known pigment cell trophic factors. (**G**) Schematic representation of pigment cell microenvironment, greatly expanded along its deep-to-superficial axis to better illustrate organization of the very flat pigment and hypodermal cells. Xanthophore location inferred from ***Hessle et al., 2013***. (**H**) Potential regulatory linkages between hypodermis and pigment cell types, inferred from

*Figure 8 continued on next page*

*Figure 8 continued*

expression (black arrows). Previously documented interactions among the pigment cells represented by gray arrows. Scale bars, 100 μm (**A**), 10 μm (**D**, **E**).

The online version of this article includes the following figure supplement(s) for figure 8:

**Figure supplement 1.** Hypodermis supports pigment cells.

**Figure supplement 2.** Csf1 requirements for xanthophore pigmentation.

various dermal cell types and pigment cells relative to wild-type controls in our sci-RNA-seq dataset and found marked deficits in hypodermal cells, as well as iridophores, melanophores, and xanthophores, whereas the numbers of dermal mesenchyme cells were relatively unchanged between genotypes (*Figure 7C*).

Based on these findings, we predicted that hypodermis would be malformed or missing in live bnc2 mutants. To test this prediction and validate inferences from the single-cell transcriptome data, we imaged the dermis of live fish expressing ET37:EGFP (*Parinov et al., 2004*; *Aman et al., 2021*). In wild-type controls, the hypodermis appeared as a thin, nearly confluent cell layer deep in the dermis, whereas in bnc2 mutants no such layer was apparent; instead, the deepest dermis appeared to contain only dermal mesenchyme (*Figure 8D*, *Figure 8—figure supplement 1A*, *Figure 1—figure supplement 1B*). Consistent with the hypodermis having roles in supporting pigment cells, we observed iridophores, expressing pnp4a:palm-mCherry and melanophores, expressing tyrp1b:palm-mCherry in close proximity to ET37:EGFP+ hypodermal cells in wild-type skin (*Figure 8E*; *McMenamin et al., 2014*; *Lewis et al., 2019*). It is possible that hypodermal cells physically persist in bnc2 mutants but have sufficiently altered transcriptional profiles such that they no longer cluster together with wild-type hypodermal cells or express the ET37:EGFP transgene. Nevertheless, these analyses suggest that ET37:EGFP+ hypodermal cells likely play a role in pigment pattern development.

Given the close contact of wild-type melanophores and iridophores with hypodermal cells, together with loss of hypodermis and pigment cells in bnc2 mutant fish, we hypothesized that factors important for pigment cell development and pattern formation are provided particularly by hypodermal cells, as opposed to other non-pigment cells of the local tissue environment (i.e. non-hypodermal stromal cells, developing SFCs, or superficially located muscle progenitor cells, also present in our dataset). To test this idea, we examined the expression of genes encoding signaling ligands and adhesion factors implicated previously in pigment pattern development.

These analyses revealed that hypodermal cells are likely to be an important driver of iridophore development as they were the principal cells to express endothelin 3b (edn3b) (*Figure 8F*, *Figure 8—figure supplement 1B*), encoding a secreted protein that is processed to an active 21 amino acid peptide required by iridophores. Diminished expression of edn3b is associated with reduced numbers of iridophores and attenuated interstripes and stripes both in zebrafish mutants and in the naturally occurring pattern of the zebrafish relative, *Danio nigrofasciatus* (*Spiewak et al., 2018*). Edn3 acts directly on iridophores but only indirectly on melanophores through an interaction between these cells and Edn-dependent iridophores (*Parichy et al., 2000*; *Krauss et al., 2014*; *Spiewak et al., 2018*).

More direct roles for hypodermal cells in regulating melanophore development were suggested by expression of other factors, including *agouti signaling protein 1* (*asip1*), which encodes a secreted protein that represses melanophore differentiation in ventral regions of the flank (*Cal et al., 2019*), though our dataset does not allow us to segment cells spatially. Interestingly, *kit ligand a* (*kitlga*), encoding a melanogenic factor that promotes survival, migration, and differentiation, was expressed at only moderate levels by hypodermal cells at this stage, despite its broad expression in the much simpler skin of embryos and early larvae (*Hultman et al., 2007*; *Brock et al., 2019*; *Dooley et al., 2013*). Nevertheless, *kitlga* was expressed at higher levels by melanophores themselves, whereas junctional adhesion molecule 3b (jam3b), encoding a 2-immunoglobulin-like domain adhesion receptor, was expressed by both hypodermal cells and melanophores. Jam3b mediates homophilic and heterophilic adhesive interactions, and is required autonomously by melanophores for an adherent phenotype; a Jam3b fusion protein accumulates at sites of overlap between mature melanophores (*Ebnet, 2017*; *Eom et al., 2017*). In the absence of Jam3b, melanophores tend to be hypopigmented, fail to acquire their mature, well-spread morphology and orderly arrangement, and many die. Together

with prior studies, our observations suggest a model in which Jam3b facilitates interactions between immature melanophores and hypodermis, with subsequent Jam3b-mediated interactions between melanophores facilitating Kitlga-dependent maturation and survival (*Figure 8H*). That a second factor likely repressive for melanogenesis, asip2b, was expressed by xanthophores further suggests a mechanism for preventing differentiation of new melanophores within the interstripe.

Our analyses point to a role for hypodermal cells in regulating xanthophore differentiation as well. Xanthophores require signaling through colony-stimulating factor-1 receptor-a (Csf1ra) for migration, survival, and differentiation (*Parichy and Turner, 2003*; *Patterson and Parichy, 2013*). One source of *Csf1* is iridophores, which express *csf1a* (*Figure 8F*): in *bnc2* mutants, xanthophores are tightly associated with residual iridophores. Nevertheless, xanthophores eventually cover the flank of *bnc2* mutants, as well as other iridophore-deficient mutants in which xanthophore development is delayed. This recovery presumably reflects expression of *csf1b*, encoding a ligand that is similarly potent to *Csf1a* in its ability to induce xanthophore differentiation, by a previously undefined population of cells in the skin (*Patterson and Parichy, 2013*). Our sci-RNA-seq dataset shows hypodermal cells to be the presumptive major source of *Csf1b*, though its transcripts were detected at lower levels in other dermal mesenchyme, superficial pre-SFCs, and other cell types as well (*Figure 8F–H*, *Figure 8— figure supplement 1C*). To test the requirements for *Csf1* genes in adult pigmentation, we generated alleles having premature termination codons. As late juveniles (~16 SSL), fish homozygous for *csf1a* mutations had overtly normal pigment patterns on the trunk but less regular patterns of pigment cells on the fins as compared to wild-type (*Figure 8—figure supplement 2*). By contrast, fish homozygous for a *csf1b* mutation were deficient for pigmented xanthophores, evident particularly on the dorsum, which lacks *csf1a*-expressing iridophores in the hypodermis, though some xanthophores persisted along scale margins where a few iridophores occur (e.g. Supplementary Figure 4 of *Guzman et al., 2013*). Fish doubly homozygous for *csf1a* and *csf1b* mutations lacked virtually all xanthophores, recapitulating the phenotype of *csf1ra* mutants. These observations support a model in which hypodermally derived *Csf1b* promotes xanthophore differentiation during normal development, and can substitute for iridophore-derived *Csf1a* in backgrounds deficient for iridophores; we presume that eventual recovery of xanthophores in *bnc2* mutants deficient for both iridophores and hypodermis reflects residual *Csf1b* availability from other dermal cell types.

Finally, genes that affect each of the major classes of pigment cell were expressed by hypodermal cells, yet *bnc2* mutants are particularly deficient for iridophores at early stages (*Patterson and Parichy, 2013*). We therefore predicted that transcriptomic states of iridophores would be more severely affected by loss of *bnc2* than transcriptomic states of melanophores or xanthophores. Consistent with this idea and a greater impact of *bnc2* loss on iridophores than other pigment cells, we found more differentially expressed genes in iridophores than xanthophores or melanophores (~1000 cells of each cell type) (*Figure 8—figure supplement 1D*). This disproportionality may be a consequence of the markedly fewer hypodermal cells and attendant loss of Edn3b or other iridogenic signals. Alternatively, *bnc2* may have more pronounced activities within iridophores, as these cells express *bnc2* and at levels greater than melanophores or xanthophores (*Figure 8F*), in addition to its non-autonomous functions in pattern formation through the hypodermis (*Lang et al., 2009*). Though further manipulative analyses will be needed to test these several interactions, our analyses of gene expression and cell type abundance identify hypodermal cells as a key source of factors permissive, and possibly instructive, for adult interstripe and stripe development.

## Discussion

Skin is a large, heterogeneous and biomedically important organ, and the skin of zebrafish is a useful system in which to elucidate mechanisms of skin patterning and morphogenesis. We have generated a minimally biased single-cell resolution transcriptional atlas of zebrafish skin at a key stage during the larva-to-adult transition, during squamation and pigment patterning. These data include transcriptomes for all major epidermal and dermal skin cell types in addition to numerous skin-associated cell types including pigment cells.

Zebrafish skin is endowed with an array of elasmoid scales, thin plates of calcified extracellular material deposited in the skin from dermal papillae that aggregate at the interface of dermis and epidermis. Calcified skin appendages in extant fish are diverse, encompassing various and sundry spines, plates, odontodes, and scales. These forms are composed of extracellular matrices that range

from among the hardest material in biology to some of the most flexible (*Sire and Huysseune, 2003*). We systematically assessed the expression of genes encoding non-collagen calcified matrix proteins throughout the skin during squamation, leading to the discovery of a transcriptionally distinct population of basal epidermal cells that express EMP transcripts, likely corresponding to epidermal secretory cells proposed to participate in scale matrix formation based on ultrastructure (*Sire et al., 1997b*). These cells also express *dlx3a, dlx4a, runx2b,* and *msx2a* but not *sp7,* a transcription factor suite that is shared with ameloblasts that form tooth enamel. While these transcription factors are not exclusive to ameloblasts and have been reported in osteoblasts and odontoblasts, in addition to cell types that do not produce calcified matrix, such as neurons, their co-expression along with EMP-encoding transcripts in basal epidermal cells is consistent with a common origin of ameloblasts and the EMP+ epidermal cells reported here. One alternative hypothesis is that co-expression of these gene products arose convergently and can be explained by mechanistic linkages among them. Future work aimed at functionally dissecting the regulatory mechanisms that govern EMP gene expression in a variety of organisms may clarify these issues either by providing evidence of additional commonalities, supporting a shared ancestor, or by revealing diverse, lineage-specific regulatory architectures, supporting convergent evolution of superficial enamel deposition in teeth and fish skin appendages.

Additionally, the complement of genes encoding matrix proteins expressed by dermal SFC suggests that although these cells may share fundamental regulatory machinery with mammalian osteoblasts and odontoblasts, including regulation by *Runx2* and *Sp7* transcription factors, they likely produce a derived form of calcified matrix, fibrillary plate elasmoidin, which is distinct from bone or dentin. Independent patterning of epidermal EMP-expressing cells and dermal SFCs might underlie some of the morphological diversity among fish skin appendages. For example, it is possible that hard spines, as in pufferfish and armored catfish, are formed by patterned expression of epidermal EMP gene products. Manipulation of matrix protein expression in zebrafish SFC and EMP+ epidermal cells may provide a powerful new system to elucidate general mechanisms of biomineralization and genetic encoding of material properties that could, in turn, inform future research in clinical dentistry.

Scales develop from dermal papillae that form under the epidermis. The regulatory underpinnings of scale papillae patterning and morphogenesis depend on reciprocal epithelial–mesenchymal signaling interactions, including contributions of Eda-A-Edar-NF-κB signaling, that are widely conserved across vertebrate skin appendages (*Cui and Schlessinger, 2006*; *Harris et al., 2008*; *Aman et al., 2018*). Analysis of scale development therefore affords a relatively accessible approach to understanding epithelial mesenchymal signaling interactions that underlie dermal morphogenesis. To this end, we generated and analyzed single-cell transcriptomes for two scaleless conditions, *eda* mutants and hypoTH fish (*Harris et al., 2008*; *McMenamin et al., 2014*). Eda is a paracrine factor that binds receptors expressed in the epidermis and thyroid hormone is an endocrine factor with potential to regulate transcription in any cell (*Cui and Schlessinger, 2006*; *Braasch et al., 2016*; *Aman et al., 2018*). Despite widely different spatial ranges over which signals are transmitted, our transcriptomic analysis suggests that both molecules regulate transcription of signaling ligands in basal epidermal cells that ultimately affect dermal morphogenesis. We further showed that Eda signaling indirectly regulates SFC differentiation by triggering expression of Fgf ligands. TH is implicated in human dermatopathies; myxedema, characterized by dry, waxy skin, is clinically synonymous with hypothyroidism (*Safer, 2011*). Yet we know remarkably little about cutaneous transcriptional targets of TH. Our analyses show that genes encoding PDGFα ligands are transcriptionally regulated by TH and can themselves regulate dermal–epidermal morphogenesis. This finding may be of relevance to understanding and potentially treating skin conditions associated with treatment-resistant TH insensitivity. Indeed, PDGFα and TH gain- and loss-of-function studies in mouse yield similar hair cycle phenotypes (*Safer et al., 2001*; *Tomita et al., 2006*; *Contreras-Jurado et al., 2015*; *González et al., 2017*).

The eponymous striped pattern of zebrafish arises from neural crest derived pigment cells that reside deep within the skin, beneath the concurrently forming scales (*Le Guellec et al., 2004*; *Hessle et al., 2013*), and depends on interactions among all three pigment cell types (*Patterson and Parichy, 2019*). Although much has been learned about stripe pattern formation from analyses of mutants lacking one or more pigment cell types, much less is known about how pigment cells integrate into the skin microenvironment. Analyses of genetic mosaics have hinted at an important role for skin cells (*Lang et al., 2009*; *Krauss et al., 2014*; *Patterson et al., 2014*; *Eom et al., 2017*); still, this aspect of pigment patterning remains largely unexplored empirically or computationally (*Volkening*

and Sandstede, 2015; Watanabe and Kondo, 2015; Volkening and Sandstede, 2018; Owen et al., 2020). Using single-cell transcriptomics and live imaging of wild-type and *bnc2* mutant fish, we have identified a discrete population of dermal cells express genes that regulate differentiation and morphogenesis of pigment cells, including *edn3b*, which is required for iridophore population expansion.

Our analyses have focused on scales and pigmentation; however, zebrafish skin is a fruitful study system for many areas of biology including regeneration and wound healing (Richardson et al., 2016; Cox et al., 2018; Iwasaki et al., 2018; Morris et al., 2018; Pfalzgraff et al., 2018); innate immunity (Lü et al., 2015; Wurster et al., 2021), stem cell regulation (Lee et al., 2014; Chen et al., 2016; Brock et al., 2019), sensory physiology and developmental neuroscience (Rasmussen et al., 2015; Rasmussen et al., 2018; Peloggia et al., 2021), and human disease modeling (Feitosa et al., 2011; Li et al., 2011a). Additionally, considerable insight into general mechanisms of development can emerge from comparing developmental mechanisms across species or between organ system. We expect the transcriptomic data presented here will help in identifying useful markers for cross-species comparisons, enabling a deeper understanding of the molecular and cellular bases of phenotypic evolution.

## Materials and methods

### Key resources table

| Reagent type (species) or resource | Designation | Source or reference | Identifiers | Additional information |
|---|---|---|---|---|
| Strain, strain background (*Danio rerio*) | sp7:EGFP; Tg(sp7:EGFP)b1212 | Gift; PMID:20506187 | RRID:ZDB-ALT-100402-1 | |
| Strain, strain background (*D. rerio*) | ET37; Et(krt4:EGFP)sqet37 | Gift; PMID:15366023 | RRID:ZDB-ALT-070702-16 | |
| Strain, strain background (*D. rerio*) | Tg(sp7:nEOS)vp46rTg | Previous study; PMID:34089732 | | |
| Strain, strain background (*D. rerio*) | (bnc2)utr16e1 | Previous study; PMID:19956727 | RRID:ZDB-FISH-150901-26677 | |
| Strain, strain background (*D. rerio*) | (eda)dt1261 | Gift; PMID:18833299 | RRID:ZDB-ALT-090324-1 | |
| Strain, strain background (*D. rerio*) | Tg(tg:nVenus-v2a-nfnB)wprt8Tg | Previous study; PMID:25170046 | RRID:ZDB-ALT-141218-1 | |
| Strain, strain background (*D. rerio*) | Tg(pnp4a:palm-mCherry)wprt10Tg | Previous study; PMID:31138706 | RRID:ZDB-ALT-200507-3 | |
| Strain, strain background (*D. rerio*) | csf1a | This study | | Available from Pavan Lab, NHGRI, Bethesda MD, USA |
| Strain, strain background (*D. rerio*) | csf1b | This study | | Available from Pavan Lab, NHGRI, Bethesda MD, USA |
| Recombinant DNA reagent | fgf20a-2A-mCherry;hsp70l:fgf20a-2A-nls-mCherry | This study | NA | Available from Parichy Lab, UVA, Charlottesville VA, USA |
| Recombinant DNA reagent | pdgfaa-2A-mCherry;hsp70l:pdgfaa-2A-nls-mCherry | This study | NA | Available from Parichy Lab, UVA, Charlottesville VA, USA |
| Antibody | Anti-Dig-AP, Fab fragments (sheep monoclonal) | MilliporeSigma | Roche; Cat# 11093274910; RRID:AB_514497 | Used at 1:5000 |
| Software, algorithm | GraphPad Prism | GraphPad | NA | |
| Chemical compound, drug | Alizarin-Red-S; ARS | MilliporeSigma | SKU_A5533 Sigma-Aldrich | |

### Zebrafish lines and husbandry

Fish were maintained in the WT(ABb) background at 28.5°C. Lines used were Tg(sp7:EGFP)[b1212] abbreviated *sp7:EGFP* (DeLaurier et al., 2010), Et(krt4:EGFP)[sqet37] abbreviated *ET37:EGFP* (Parinov et al., 2004), Tg(sp7:nEOS)[vp46rTg] (Aman et al., 2021), bnc2[utr16e1] (Lang et al., 2009), eda[dt1261] (Harris et al.,

2008), *Tg(tg:nVenus-v2a-nfnB)*[wprt8Tg] abbreviated *tg:Venus-NTR*, *Tg(tyrp1b:palm-mCherry)*[wprt11Tg] (*McMenamin et al., 2014*), and *Tg(pnp4a:palm-mCherry)*[wprt10Tg]. Thyroid ablation and rearing of hypoTH fish were done as previously described (*McMenamin et al., 2014*). *csf1a* and *csf1b* mutant fish were generated by injecting CRISPR/Cas9 reagents (PNAbio) including synthetic single-guide RNA targeting the genomic sequences (*csf1a*: 5'-GCGGCATTCCCTCACATAC; *csf1b*: 5'-GGCATGTT TGCAAGGACCG) into zygotes, selecting phenotypic F0 animals, and repeat outcrossing to generate F2 families (*Hwang et al., 2013*). Recovered alleles contained premature termination codons owing to frame shift mutations (*csf1a*: 10 bp insertion; *csf1b*: 2 or 5 bp deletions having phenotypes indistinguishable from one another).

## Imaging

Alizarin-Red-S vital dye, MS-222 anesthesia, and mounting for microscopy were performed as previously described (*Aman et al., 2021*). Images in *Figure 1—figure supplement 1B*, *Figure 2C and D*, *Figure 4C*, *Figure 5E*, and *Figure 8A, D and E* were acquired on a Zeiss LSM880 in fast Airyscan mode. Images in *Figure 2—figure supplement 1A* and *Figure 4A* were acquired on a Zeiss LSM880 in conventional confocal mode. Images in *Figure 1A*, *Figure 1—figure supplement 1C*, *Figure 3C and E*, *Figure 5D*, *Figure 6E*, and *Figure 6—figure supplement 1A and B* were acquired on a Zeiss Observer equipped with Yokogawa CSU-X1 spinning disc. Images in *Figure 8—figure supplement 2* were acquired on a Zeiss SteREO Discovery V12 stereomicroscope. Orthogonal views were produced using FIJI (*Schindelin et al., 2012*). Brightness and contrast were adjusted in Adobe Photoshop, and nonlinear gamma adjustments were applied to images when necessary to highlight relevant cell types. Photoconversion of nuclear EOS was done using a 405 nm laser on a Zeiss LSM800.

## mRNA in situ hybridization

All in situ hybridization probe templates were amplified using Primestar-GXL (Takara) from cDNA prepared with SSIII (ThermoFisher) with the following primers: ambn 5'-TGATGATCGTGTGCTTTCTT GCTG, 5'-aaaaTAATACGACTCACTATAGCATTTTGCCCCTGTTGTGGTCTTG; itga5 5'-AGGAAGGA AGTGTACATGGGTGA, 5'-aaaaTAATACGACTCACTATAGgatccagttttgtcccagatgac; itgb3b 5'-TGGA CCTGTCCTACTCCATGAAT, 5'-aaaaTAATACGACTCACTATAGacactgtcttttttagcgctgtcc; col10a1a 5'-gaacccaagtatgccgatttgacc, 5'-aaaaTAATACGACTCACTATAGtgttttgatgtgatgtggatgggt; col10a1b 5'-gcttagcttcagaaaATGGACCTCA, 5'-aaaaTAATACGACTCACTATAGTGGTTGTCCCTTTTCACCTGGA TA; tcf7 5'-CCAACAAGGTGTCGGTGGT, 5'-aaaaTAATACGACTCACTATAGACCAGTCCGTCTGttggt tcag; jag1a 5'-CCCTTGACCAAACAAATGACAA, 5'-aaaaTAATACGACTCACTATAGGCTGTGTTTT CTTCAGGTGTGG. chad 5'-AGACCAAACATCCAGACAGCAA, 5'-aaaaTAATACGACTCACTATAGGC AATTGCATCATCCTTCACAT. In situ hybridization probes and tissue were prepared as described previously (*Quigley et al., 2004*), with hybridization and post-hybridization washes performed on a BioLane HTI 16 Vx platform (Intavis Bioanalytical Instruments) and post-staining vibratome sectioning in some cases as described (*Aman et al., 2018*).

## Heatshock transgene cloning and expression

Full-length coding sequences were amplified from SSIII cDNA (Thermo Fisher) using Primestar-GXL polymerase (Takara) and the following primers: *fgf20a* 5'-AAGCAGGCTCACCATGGGTGCAGTCGGC GA; 5'- GACTGCACCCATGGTGAGCCTGCTTTTTTGTACAAACTTGG; *pdgfaa* 5'- GCAGATATAAGG TGCGCCAGCGTCACCCA, 5'-CGCGGTTCTCATGGTGAGCCTGCTTTTTTGTACAAACTTGG. Coding sequences were cloned into a hsp70l heatshock misexpression vector from *Aman et al., 2018* using NEBuilder HiFi DNA Assembly Master Mix (NEB). Zygotes were injected and raised to 8.5 SSL and given 6 × 1 hr 41°C heatshocks per day for 7 d in a modified Aquaneering rack. Only individuals with epidermal basal cell expression were selected for analysis.

## Tissue dissection and storage

Fish were staged according to *Parichy et al., 2009*, and 9.6 SSL individuals were selected for dissection, euthanized with MS-222, and processed immediately. Following removal of the head and fins, skins of wild-type controls, *eda* mutant homozygotes, *bnc2* homozygotes, and hypoTH fish in an *sp7:EGFP* transgenic background zebrafish were removed with forceps and immediately flash frozen in liquid nitrogen, then stored at –80°C prior to isolation of nuclei (n = 300 fish skins total).

## Nuclei isolation and sci-RNA-seq2 library preparation

Separately for each background, frozen skins (n = ~60) were thawed over ice in cold lysis buffer (10 mM Tris-HCl, pH 7.4, 10 mM NaCl, 3 mM MgCl$_2$, 0.1% IGEPAL CA-630) (*Cao et al., 2019*) supplemented with 5% Superase RNA Inhibitor and minced with a razorblade until no visible pieces remained (<1 min). The cell suspension was then pipetted a few times and put through a 50 µM filter into 10 ml of fixation buffer (5% paraformaldehyde, 1.25× PBS) (*Srivatsan et al., 2020*). Nuclei were fixed on ice for 15 min then centrifuged at 700 × *g* for 10 min. Fixed nuclei were subsequently rinsed twice with 1 ml of nuclei resuspension buffer (NSB: 10 mM Tris-HCl, pH 7.4, 10 mM NaCl, 3 mM MgCl$_2$, 1% Superase RNA Inhibitor, 1% 0.2 mg/ml Ultrapure BSA), spun down at 750 × *g* for 6 min, and incubated in 400 µl of permeabilization buffer (NSB+ 0.25% Triton-X) for 3 min on ice. Permeabilized nuclei were spun down, resuspended in 400 µl of NSB, and sonicated on 'low' for 12 s. Following sonication, nuclei were spun down once more, resuspended in 400 µl of NSB, and nuclei from each sample were DAPI-stained and counted on a hemocytometer. Sci-RNA-seq2 libraries were then prepared as previously described (*Cao et al., 2017*). Briefly, 1200 nuclei in 2 µl of NSB and 0.25 µl of 10 mM dNTP mix (Thermo Fisher Scientific, Cat#R0193) were distributed into each well of 12 96-well plates—4 per background (LoBind Eppendorf). Then, 1 µl of uniquely indexed oligo-dT (25 µM) (*Cao et al., 2017*) was added to every well, incubated at 55°C for 5 min and placed on ice. Also, 1.75 µl of reverse transcription mix (1 µl of Superscript IV first-strand buffer, 0.25 µl of 100 mM DTT, 0.25 µl of Superscript IV, and 0.25 µl of RNAseOUT recombinant ribonuclease inhibitor) was then added to each well and plates incubated at 55°C for 10 min and placed on ice. Wells were pooled, spun down, and resuspended in 500 µl NSB and transferred to a flow cytometry tube through a 0.35 µm filter cap; DAPI was added to a final concentration of 3 µM. Pooled nuclei were then sorted on a FACS Aria II cell sorter (BD) at 300 cells per well into 96-well LoBind plates containing 5 µl of EB buffer (QIAGEN). After sorting, 0.75 µl of second strand mix (0.5 µl of mRNA second-strand synthesis buffer and 0.25 µl of mRNA second-strand synthesis enzyme, New England Biolabs) were added to each well, second-strand synthesis performed at 16°C for 150 min. Tagmentation was performed by addition of 5.75 µl of tagmentation mix (0.01 µl of an N7-only TDE1 enzyme [in-house] in 5.7 µl 2× Nextera TD buffer, Illumina) per well and plates incubated for 5 min at 55°C. Reaction was terminated by addition of 12 µl of DNA binding buffer (Zymo) and incubated for 5 min at room temperature. Then, 36 µl of Ampure XP beads were added to every well, DNA purified using the standard Ampure XP clean-up protocol (Beckman Coulter) eluting with 17 µl of EB buffer and DNA transferred to a new 96-well LoBind plate. For PCR, 2 µl of indexed P5, 2 µl of indexed P7 (*Cao et al., 2017*), and 20 µl of NEBNext High-Fidelity master mix (New England Biolabs) were added to each well and PCR performed as follows: 75°C for 3 min, 98°C for 30 s, and 19 cycles of 98°C for 10 s, 66°C for 30 s, and 72°C for 1 min followed by a final extension at 72°C for 5 min. After PCR, all wells were pooled, concentrated using a DNA clean and concentrator kit (Zymo), and purified via an additional 0.8× Ampure XP cleanup. Final library concentrations were determined by Qubit (Invitrogen), libraries visualized using a TapeStation D1000 DNA Screen tape (Agilent), and libraries sequenced on a Nextseq 500 (Illumina) using a high-output 75 cycle kit (read 1: 18 cycles; read 2: 52 cycles; index 1: 10 cycles; and index 2: 10 cycles).

## Preprocessing of sequencing data

Sequencing runs were demultiplexed using bcl2fastq v.2.18 and expected PCR barcode combinations. The backbone computational pipeline for read processing was previously published (*Cao et al., 2017*). Following assignment of RT indices, reads were trimmed using trim-galore and mapped to the zebrafish transcriptome (GRCz11 with extended 3' UTRs) (*Saunders et al., 2019*) using the STAR aligner (*Dobin et al., 2013*). Reads were then filtered for alignment quality, and duplicates were removed. Non-duplicate genes were assigned to genes using bedtools (*Quinlan and Hall, 2010*) to intersect with an annotated gene model. Cell barcodes were considered to represent a real cell if the number of UMIs was greater than 600, a number chosen based on a user-defined threshold on the knee plot. Cells with greater than 6000 UMIs were also discarded as likely multiplets. Reads from cells that passed the UMI thresholds were aggregated into a count matrix and then loaded and saved as a CDS object for downstream analysis with monocle3 (*Cao et al., 2019*).

## Dimensionality reduction, alignment, and background correction

The wild-type-only (n = 35,114) and all-background (wild-type, *eda* mutant, *bnc2* mutant, hypoTH; n = 144,466) CDS objects were processed separately. Cells were assigned to their background by matching recovered RT barcode information to the original plate loadings. For each dataset, the standard monocle3 processing workflow was followed (estimate_size_factors(), detect_genes(), preprocess_cds()) and the top 50 PCs were retained, and PCA was calculated using all genes as input. A few corrections were then made on the original PCA matrix. First, to account for possible cytoplasmic RNAs in the supernatant of each sample that could contribute to 'background' in the resulting transcripts assigned to individual cells, we performed a sample-specific background correction as previously described (*Packer et al., 2019*). Briefly, the background distribution of RNA from was calculated from 'cells' that had less than 15 UMIs, and we used this to compute a 'background loading.' Next, a linear regression model was fit using these background loadings (real cell PCA matrix ~ cell background loadings), and its residuals were considered the 'background corrected PCA matrix.' This corrected PCA matrix was then subject to Mutual Nearest Neighbor (MNN) alignment (*Haghverdi et al., 2018*) by sample using the 'align_cds' function in monocle3. The background-corrected, MNN-aligned PCA matrix was then used as input for Uniform Manifold Approximation and Projection (UMAP) (*Becht et al., 2018*) dimensionality reduction using the 'reduce_dimension' function and default settings (except umap.min_dist = 0.15, umap.n_neighbors = 20L). Clustering was performed with 'cluster_cells' (wild-type resolution = 2e-4; all-background resolution = 1e-4), which uses the Leiden community detection algorithm (*Traag et al., 2019*). Clustering resolution was selected manually based on clear distinction of non-adjacent groups of cells and a reasonable recovery of overall UMAP structure.

## Cell type classification and trajectory analysis

For each cluster in the wild-type dataset, the most specific genes were calculated using the 'top_markers' function. These genes were sorted by specificity, and clusters were annotated by comparing genes to published studies and in situ hybridization databases. We assigned 43 clusters to 33 unique cell types and one 'unknown' group when the cell type was not able to be determined based on gene expression. To annotate cells from the *eda* mutant, *bnc2* mutant, and hypoTH backgrounds, we built a marker-free cell type classifier with the wild-type cell annotations using Garnett (*Pliner et al., 2019*) and applied it to the remaining cells. Trajectory analysis was performed on a subsetted and reprocessed set of cells from the wild-type dermis. These cells form the developing scales, and we chose a single stage that contained scale-forming cells along the entire developmental trajectory. After repeating dimensionality reduction, we applied the monocle3 function 'learn_graph' and 'order_cells' to root the graph and calculate 'pseudotime' values for each cell that increased as a function of principal graph distance from the root.

## Analysis of cell type abundance differences

To compute cell type abundance differences across genotypes, cell counts for each type were normalized by the sample's size factor (the total cell counts from each sample divided by the geometric mean of all sample's total cell counts). The abundance difference was then calculated as $\log_2$(normalized query cell count/normalized reference cell count) for each cell type relative to wild-type for all backgrounds.

## Differential expression analysis

Differentially expressed genes were computed by fitting the size-factor normalized UMI counts for each gene from each individual nucleus with a generalized linear model using the 'fit_models' function in monocle3. To fit the regression model for each background's effect on each gene in each cell type, we first selected the pair-wise backgrounds and cells that were relevant for the model (i.e. basal cells from wild-type and *eda*). Then we filtered for genes expressed in at least 10 cells and used background as the model formula (model_formula_str = '~genotype') and extracted the coefficient table, p-values, and multiple testing corrected q-values with the 'coefficient_table' function. Genes were considered significantly background-dependent differentially expressed (DEGs) if their q-value was less than 0.05. For analysis of specific pathways, we used *D. rerio* gene–pathway associations from WikiPathways (*Martens et al., 2021*).

## Acknowledgements

This work was supported by NIH R35 GM122471 and NIH R01 AR078320 (DMP) as well as NIH U54 HL145611, NIH UM1 HG011586, NIH R01 HG010632 to CT and the Paul G Allen Frontiers foundation (Allen Discovery Center) (CT). Thanks to Amber Schwindling, other Parichy lab members, and Raman Sood for assistance and oversight of fish care.

## Additional information

### Funding

| Funder | Grant reference number | Author |
|---|---|---|
| National Institute of General Medical Sciences | R35 GM122471 | David M Parichy |
| National Institute of Arthritis and Musculoskeletal and Skin Diseases | R01 AR078320 | David M Parichy |
| National Heart, Lung, and Blood Institute | U54 HL145611 | Cole Trapnell |
| National Human Genome Research Institute | UM1 HG011586 | Cole Trapnell |
| National Human Genome Research Institute | R01 HG010632 | Cole Trapnell |
| Paul G. Allen Frontiers Group | | Cole Trapnell |

The funders had no role in study design, data collection and interpretation, or the decision to submit the work for publication.

### Author contributions

Andrew J Aman, Conceptualization, Formal analysis, Supervision, Funding acquisition, Validation, Investigation, Visualization, Methodology, Writing – original draft, Project administration, Writing – review and editing; Lauren M Saunders, Conceptualization, Data curation, Formal analysis, Validation, Investigation, Visualization, Methodology, Writing – original draft, Writing – review and editing; August A Carr, Investigation; Sanjay Srivatasan, Methodology; Colten Eberhard, Blake Carrington, Dawn Watkins-Chow, William J Pavan, David M Parichy, Resources; Cole Trapnell, Funding acquisition, Project administration

### Author ORCIDs

Andrew J Aman (ORCID) https://orcid.org/0009-0003-3311-4661
Lauren M Saunders (ORCID) https://orcid.org/0000-0003-4377-4252
Colten Eberhard (ORCID) https://orcid.org/0000-0001-7292-7313
David M Parichy (ORCID) https://orcid.org/0000-0003-2771-6095

### Ethics

This study was performed in strict accordance with the recommendations in the Guide for the Care and Use of Laboratory Animals of the National Institutes of Health. All of the animals were handled according to approved institutional animal care and use committee (IACUC) protocol (#4170) of the University of Virginia.

Reviewer #1 (Public Review): https://doi.org/10.7554/eLife.86670.4.sa1
Reviewer #2 (Public Review): https://doi.org/10.7554/eLife.86670.4.sa2
Author Response: https://doi.org/10.7554/eLife.86670.4.sa3

## Additional files

### Supplementary files

• Supplementary file 1. Published markers of skin and skin-associated cell types.

• Supplementary file 2. scRNA-seq metadata. These tables include marker genes for both assigned cell types and unsupervised clustering, as well as definitions of gene signatures and differentially expressed genes between euthyroid and hypothyroid conditions for each cell type.

• MDAR checklist

### Data availability

Sequencing data have been deposited in GEO under accession code GSE224695. Figure and analysis code is available from https://github.com/lsaund11/zfish-skin (copy archived at *Saunders, 2023*).

The following dataset was generated:

| Author(s) | Year | Dataset title | Dataset URL | Database and Identifier |
|---|---|---|---|---|
| Parichy D, Aman A, Saunders L, Trapnell C | 2023 | Transcriptomic profiling of tissue environments critical for post-embryonic patterning and morphogenesis of zebrafish skin | https://www.ncbi.nlm.nih.gov/geo/query/acc.cgi?acc=GSE224695 | NCBI Gene Expression Omnibus, GSE224695 |

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
