## [Editor Report · eLife assessment]

This study provides a clearly presented and thoughtfully analyzed single cell-resolution dataset of gene expression in wildtype and mutant zebrafish skin. These data are used by the authors to develop and test hypotheses about cell lineage relationships and signaling interactions between cell types in the skin, allowing them to identify roles for several signaling pathways and the hypodermis in scale and pigment cell development. These findings constitute a **fundamental** contribution to the field, and the rigor of the analyses make this manuscript **compelling**.

---

## [Referee Report · Reviewer #1 (Public Review)]

In their study, Aman et al. utilized single cell transcriptome analysis to investigate wild-type and mutant zebrafish skin tissues during the post-embryonic growth period. They identified new epidermal cell types, such as ameloblasts, and shed light on the effects of TH on skin morphogenesis. Additionally, they revealed the important role of the hypodermis in supporting pigment cells and adult stripe formation. Overall, I find their figures to be of high quality, their analyses to be appropriate and compelling, and their major claims to be well-supported by additional experiments. Therefore, this study will be an important contribution to the field of vertebrate skin research.

---

## [Referee Report · Reviewer #2 (Public Review)]

This work describes transcriptome profiling of dissected skin of zebrafish at post-embryonic stages, at a time when adult structures and patterns are forming. The authors have used the state-of-the-art combinatorial indexing RNA-seq approach to generate single cell (nucleus) resolution. The data appears robust and is coherent across the four different genotypes used by the authors.

The authors present the data in a logical and accessible manner, with appropriate reference to the anatomy. They include helpful images of the biology and schematics to illustrate their interpretations.

The datasets are then interrogated to define cell and signalling relationships between skin compartments in six diverse contexts. The hypotheses generated from the datasets are then tested experimentally. Overall, the experiments are appropriate and rigorously performed. They ask very interesting questions of interactions in the skin and identify novel and specific mechanisms. They validate these well.

The authors use their datasets to define lineage relationships in the dermal scales and also in the epidermis. They show that circumferential pre-scale forming cells are precursors of focal scale forming cells while there appeared a more discontinuous relationship between lineages in the epidermis.

The authors present transcriptome evidence for enamel deposition function in epidermal subdomains. This is convincingly confirmed with an ameloblastin in situ. They further demonstrate distinct expression of SCPP and collagen genes in the SFC regions.

The authors then demonstrate that Eda and TH signalling to the basal epidermal cells generates FGF and PDGF ligands to signal to surrounding mesenchyme, regulating SFC differentiation and dermal stratification respectively.

Finally, they exploit RNA-seq data performed in parallel in the bnc2 mutants to identify the hypodermal cells as critical regulators of pigment patterning and define the signalling systems used.

Whilst these six interactions in the skin are disparate, the stories are unified by use of the sci-RNA-seq data to define interactions. Overall, it's an assembly of work which identifies novel and interesting cell interactions and cross-talk mechanisms.

The paper provides robust evidence of cell interrelationships in the skin undergoing morphogenesis and will be a welcome dataset for the field.

---

## [Author Response]

The following is the authors’ response to the previous reviews.

**Reviewer #3 (Recommendations For The Authors):**
In response to my comment about Col10a1 expression in the dermal SFCs (Fig 3B, I), the authors provide additional text to clarify but also state that "Col2 genes were not detected robustly". I think this comment on the absence of Col2 transcripts should be explicitly included in that paragraph as it is a reasonable and expected question given the cartilage angle the authors begin the paragraph with. Including this in no way weakens their point, rather adds clarity.

This version includes some modifications to fix typos and add a sentence in response to the concern above.